# A Hierarchical Reinforcement Learning Based Optimization Framework for Large-scale Dynamic Pickup and Delivery Problems

Yi Ma[1][*], Xiaotian Hao[1][*], Jianye Hao[12][†], Jiawen Lu[2], Xing Liu[2],
Xialiang Tong[2], Mingxuan Yuan[2], Zhigang Li[1], Jie Tang[3], Zhaopeng Meng[1]
[1]College of Intelligence and Computing, Tianjin University
{mayi,xiaotianhao, jianye.hao, scs_lzg, mengzp}@tju.edu.cn
[2]Noah's Ark Lab, Huawei, {jiawen.lu, tongxialiang, Yuan.Mingxuan}@huawei.com
[3]Tsinghua University, jietang@tsinghua.edu.cn

## Abstract

The Dynamic Pickup and Delivery Problem (DPDP) is an essential problem in the logistics domain, which is NP-hard. The objective is to dynamically schedule vehicles among multiple sites to serve the online generated orders such that the overall transportation cost could be minimized. The critical challenge of DPDP is the orders are not known a priori, i.e., the orders are dynamically generated in real-time. To address this problem, existing methods partition the overall DPDP into fixed-size sub-problems by caching online generated orders and solve each sub-problem, or on this basis to utilize the predicted future orders to optimize each sub-problem further. However, the solution quality and efficiency of these methods are unsatisfactory, especially when the problem scale is very large. In this paper, we propose a novel hierarchical optimization framework to better solve large-scale DPDPs. Specifically, we design an upper-level agent to dynamically partition the DPDP into a series of sub-problems with different scales to optimize vehicles routes towards globally better solutions. Besides, a lower-level agent is designed to efficiently solve each sub-problem by incorporating the strengths of classical operational research-based methods with reinforcement learning-based policies. To verify the effectiveness of the proposed framework, real historical data is collected from the order dispatching system of Huawei Supply Chain Business Unit and used to build a functional simulator. Extensive offline simulation and online testing conducted on the industrial order dispatching system justify the superior performance of our framework over existing baselines.

## 1 Introduction

The Dynamic Pickup and Delivery Problem (DPDP) constitutes an important family of routing problems, which generally contains three key elements: orders, goods and vehicles as shown in Figure 1. Orders are generated in real-time. Different orders contain different types and quantities of goods. A number of vehicles are scheduled to serve the orders by transporting the desired goods from different origins to different destinations. The objective of DPDP is to dynamically assign each order to the most appropriate vehicle so that the overall transportation cost (e.g., overall distances) could be minimized. DPDPs are widespread in order dispatching systems of the supply chain, express mail delivery services and elsewhere.

---

[*]Equal contribution. [†] Corresponding author.

DPDP is a complex variant of the Travelling Salesman Problem (TSP) and Vehicle Routing Problem (VRP), which are both NP-Hard combinatorial optimization problems [25]. The main difficulty of DPDP comes from the dynamically generated orders in real-time, thus the order dispatching decisions cannot be made beforehand in an offline style. Besides, compared with TSP and VRP, there exist various additional complex constraints in DPDP such as pickup and delivery constraint, Last-In-First-Out (LIFO) constraint, time window constraint, split demand constraint, etc.

**Traditional methods for DPDP**. Existing solutions for DPDP fall into two categories. The first category maintains a fixed buffer to cache the most recent generated orders and periodically dispatches all cached orders in a delayed mode. By this way, the overall dynamic problem is partitioned into a series of static sub-problems with subsets of known orders, i.e., static Pickup and Delivery Problems (PDPs). Then, operational research-based (OR) methods [22, 20], heuristic and

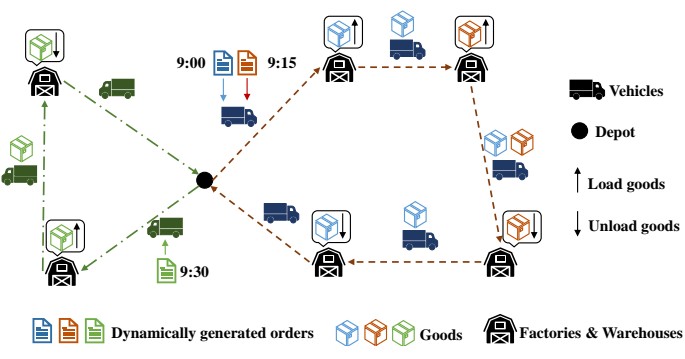

Figure 1: Demonstration of DPDP

meta-heuristic methods [7, 16, 25, 4, 5, 3, 11, 26, 23, 9] are designed to solve each sub-problem. However, myopically optimizing each static sub-problem cannot guarantee the overall dynamic problem could be optimized from a long-term perspective since the split sub-problems are not independent of each other. The main reasons are previous orders assignment results will influence (1) the number of remaining orders to be dispatched, (2) the vehicle's remaining capacity and (3) the relative positions to the following orders. To acquire better solutions, the second category methods [24, 8, 12] try to predict the distribution of future orders and take the predicted orders into consideration when computing the solution for each sub-problem. However, predicting future orders is not realistic due to the high uncertainty in the real world. Inaccurate predictions will mislead the order dispatcher and route planner, and result in poor solution quality.

**Learning-based methods for VRP**. Additionally, a common flaw of traditional OR and meta-heuristic methods is that they are computationally expensive and normally unable to obtain a desired solution within the allowable time. Besides, the design of them heavily relies on complex domain knowledge. To improve the solution computing efficiency and ease the difficulty of the algorithm design, recently, several learning-based methods are proposed [27, 1, 18, 6, 13]. These methods have demonstrated that the solution computing efficiency can be significantly improved by leveraging the generalization ability of the trained models. Besides, they could obtain solutions with competitive qualities compared with the state-of-the-art traditional methods. Although these methods mainly focus on TSPs or VRPs, of which all orders' information is known in advance and much fewer constraints are considered comparing with DPDP, learning-based methods have shown great potential to help solve large-scale DPDPs and reach superior performance.

In this paper, we propose a novel hierarchical reinforcement learning (RL) based optimization framework to solve the real-world large-scale DPDPs. Considering that order dispatching has a long-term impact on the overall optimization objective, the upper-level RL agent dynamically determines whether to wait longer at each moment for caching more future orders. In this way, the orders can be more flexibly assigned to vehicles (since each vehicle will have more candidate orders to choose) and the routes of vehicles could be optimized towards globally better solutions. The lower-level RL agent is responsible for assigning the cached orders to the most appropriate vehicles by sequentially manipulating heuristic operators to improve the solution quality iteratively. To verify the effectiveness of the framework, we collected real historical data from the order dispatching system of Huawei Supply Chain and built a simulator to simulate the order dispatching and vehicle transportation process. Further, we deployed our method on the company's Supply Chain Business Unit. Extensive offline simulation and online testing showed the superior performance of our algorithm.

Our main contributions are as follows:

- We are the first to propose a practical hierarchical RL framework to efficiently and far-sightedly compute superior solutions for the real-world large-scale DPDPs with complex constraints.
- We design a simulator using real industrial data to be the experimental benchmark to verify the proposed method, which is available here for interested researchers.
- We show that our approach considerably improves the optimization objectives compared with existing algorithms both in the offline evaluation and online testing. The ablation study indicates our approach can obtain high-quality solutions with fast running speed and has strong generalization ability.

## 2   Problem Formulation

We now give the formulation of DPDP in our logistics scenario. For the orders dynamically generated in real-time at different nodes (i.e., factories and warehouses) within a day, vehicles should be scheduled to transport the goods from pickup nodes to delivery nodes to fulfil the orders with minimal transportation cost. In our case, the objective is to minimize $K$ vehicles average travelling distances $D(K)$ of the entire DPDP:

$$\min D(K) \tag{1}$$

while meeting several constraints: Pickup and Delivery Constraint, Capacity Constraint, LIFO Constraint, Time Window Constraint, etc. Detailed constraints are shown in Appendix A.

In practice, however, some orders are destined to violate time window constraints[2]. Thus, we add it to the objective function as an associated penalty to convert the hard time window constraint to a soft one. The penalty function is defined as the *overtime* beyond the specified completion time of each order. The optimization objective is then reformulated as minimizing the weighted sum [3] of average vehicles travelling distances (kilometers) $D(K)$ and total overtime (seconds) $OT$ of all orders $C$:

$$\min D(K) + \lambda * OT(C) \tag{2}$$

Apart from the various complex constraints mentioned above, the additional difficulties of this problem mainly come from two aspects:

(1) The problem scale is very large. In practical logistic scenarios of the company, millions of products and intermediate materials are manufactured every day. As these products and materials might be used in the subsequent phases (e.g., assembling or selling), they have to be scheduled and transported between hundreds of factories and warehouses by dozens of vehicles within stringent timeline constraints, which constitutes a very large-scale and complex DPDP.

(2) Besides, as the orders are generated online in real-time, the schedule planning cannot be made aforehand in an offline style. From the oracle's point of view, i.e., when all orders of a day are known in advance, the uncertainty is eliminated and this DPDP can be formulated as a complex Mixed Integer Programming (MIP) Problem, of which the optimal solution could be obtained utilizing exact algorithms (e.g., cutting plane algorithms, branch-and-bound algorithms or modern solvers such as Gurobi[20]) [22]. However, in reality, it's impossible to know all the orders in advance, thus these approaches are not applicable.

To eliminate the uncertainties brought by the unknown orders, a practical way is to utilize a buffer to cache the most recent generated orders and periodically dispatches all cached orders in a little delayed mode. With the known orders in the cache, the static PDP can be formulated as an MIP as shown in Appendix A. We could resort to modern solvers to solve this MIP. However, even for the static PDP with very few orders, it still costs several hours to compute a feasible solution, which is beyond the acceptable limits (details are shown in Table 2 and 3). Besides, even if we could obtain the optimal solution for each *fixed split* static PDP, we still cannot guarantee the global DPDP can be optimized as these static sub-problems are not independent of each other.

---

[2]For example, goods of an order need to be transported to a far destination in a short time. Even if a vehicle is used to serve the order as soon as the order is generated, violation of time window constraints can still occur.

[3]If the overtime happens, the subsequent assembling and selling will be delayed or cancelled, resulting in great economic loss. Therefore, we set $\lambda$ to a very large number, e.g., 10000, to severely punish the overtime.

# 3 Method

## 3.1 Overall Framework

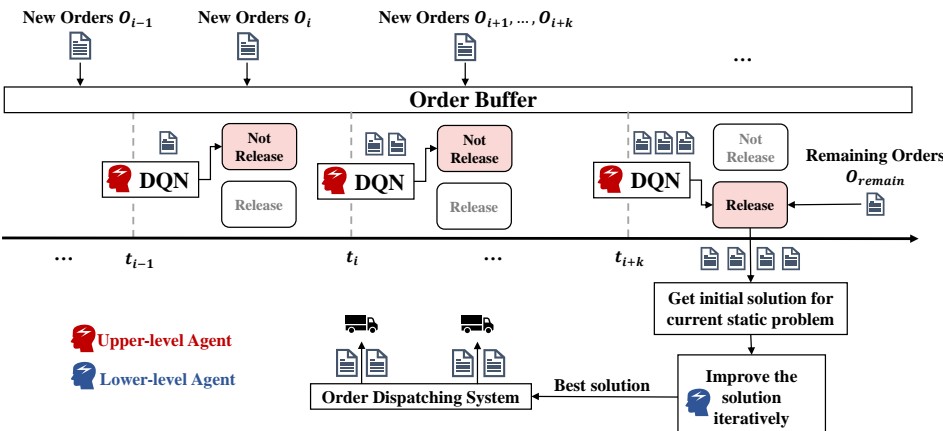

Figure 2: Hierarchical Optimization Framework

In this paper, considering the challenges mentioned above, we propose a novel hierarchical reinforcement learning based optimization framework, which contains two levels of agents. As shown in Figure 2, we maintain a *buffer* to cache the newly generated orders and periodically dispatch all cached orders at once. But instead of dispatching the cached orders of fixed numbers or predicting future orders, we design an upper-level agent to *dynamically determine whether to wait longer for caching more future orders at each moment*. Though waiting longer will postpone the dispatching and transportation of the earlier cached orders, additional future orders can be taken into account for the vehicle-order matching. In this way, each vehicle will have more candidate orders to choose, thus the overall travelling distances will be more potentially to be optimized for shorter[4]. This process could be regarded as sacrificing a little time in exchange for a precise estimation of future orders. However, waiting for too long will also increase the risk of overtime of the earlier cached orders. Thus, whether to wait longer to cache more orders at each moment will have a long-term impact on the overall dispatching results, and can be naturally modeled as a sequential decision-making problem. We model this procedure as a Markov Decision Process (MDP). Depending on whether to wait longer at each moment, the overall DPDP can be dynamically partitioned into a series of static sub-problems, each of which includes different numbers of orders. As shown in Figure 2, the generated orders are accumulated in the buffer until the upper-level agent decides to stop caching at time $t_{i+k}$. Then, the agent releases the cached orders to the lower-level agent and clears the buffer.

Given the released orders (which form a static sub-problem, i.e., a PDP), the lower-level agent is appointed to assign the orders to the most appropriate vehicles and arrange the transportation route of each vehicle, such that the transportation cost of these orders could be minimized. First, a set of basic operators are maintained, whose roles are converting one feasible solution to another. For instance, given an initial solution's route {A->B->C} with three nodes A, B and C, a typical operator is *swapping two nodes*[13], e.g., swapping A and B. After applying this operator, {A->B->C} is converted to {B->A->C}. If the travelling cost of {B->A->C} is less than {A->B->C}, the initial solution is improved. On this basis, we design the lower-level agent similar to the traditional meta-heuristic algorithms which sequentially manipulates these operators to improve the solution of each PDP. The difference is that we model the process of sequentially manipulating these operators as an MDP and incorporate RL methods to optimize the policy instead of manually designing complex rules. Finally, the best found solution is adopted by the order dispatching system to assign the orders to the vehicles and arrange their transportation routes. In the following two subsections, we will go into more details of the designed two agents.

---

[4]In Table 7 of Appendix I, we show that splitting the overall DPDP into different sub-problems with different time spans will lead to different qualities of solutions

### 3.2 Upper-level Agent

#### 3.2.1 Workflow

The workflow of the upper-level agent is described in Figure 2. We partition a day into $T = 144$ fixed time intervals, and the time span of each interval is ten minutes. Each time interval starts at time $t_{i-1}$ and ends at time $t_i$, $i \leq T$ is a positive integer. We name $t_1, ..., t_i, ..., t_T$ as *decision points*. At each decision point $t_i$, the upper-level agent decides whether to release the accumulated orders to the lower-level agent according to their overtime risk. Taking Figure 2 as an example, at decision point $t_{i-1}$, the buffer already cached some orders $O_{i-1}$. At $t_{i-1}$, the upper-level agent makes a decision and determines to wait longer and not to release $O_{i-1}$ to the lower-level agent. Thus $O_{i-1}$ are still maintained in the buffer. Thereafter, at all decision points before $t_{i+k}$, the upper-level agent makes the same decisions as at $t_{i-1}$, i.e., 'not release'. Therefore, new generated orders $\langle O_i, ..., O_{i+k} \rangle$ between $t_{i-1}$ and $t_{i+K}$ are all appended to the buffer as well. At $t_{i+k}$, the upper-level agent makes a change and determines to release all accumulated orders $\langle O_{i-1}, O_i, ..., O_{i+k} \rangle$ to the lower-level agent. At this time, all accumulated orders together with the remaining orders $O_{\text{remain}}$ (assigned to the vehicles before $t_{i-1}$ but the goods of the orders are still not loaded onto the vehicles even at $t_{i+k}$) will be released by the upper-level agent. In this way, we get a static sub-problem constituting of orders $\langle O_{i-1}, O_i, ..., O_{i+k}, O_{\text{remain}} \rangle$ for the lower-level agent. By analogy, the overall DPDP can be divided into a series of static PDPs with different scales. We model the procedure of whether to wait longer at each decision moment as an MDP described in the following subsection.

#### 3.2.2 Markov Decision Process (MDP)

**State:** The state includes the number of orders accumulated in the buffer, the number of available vehicles, the amount of time left before exceeding the time limit of each order, etc. All these features are normalized and concatenated together. Detailed descriptions are postponed to the Appendix F due to the space limitation.

**Action:** The action is a binary variable indicates whether to release orders to the lower-level agent.

**Reward:** Our ultimate goal is to minimize the optimization objective for the entire dynamic problem. Therefore, we first get the the overtime of the orders completed and the corresponding vehicle travelling distances between two consecutive decision moments $t_{i-1}$ and $t_i$ as shown in Figure 2, i.e., avg_distance $+ \lambda *$ overtime. Then we set the immediate reward of action executed at $t_{i-1}$ as $-(\text{avg\_distance} + \lambda * \text{overtime})$. By this rule, the sum of the immediate rewards forms the negative value of the total objective for the entire dynamic problem. With this reward, the overall objective for the entire dynamic problem could be optimized. In other words, we encourage the upper-level agent to optimize the overall dynamic problem from a long-term perspective when making decisions.

#### 3.2.3 Agent Model

For the upper-level agent, we use Deep Q Network (DQN) [17]. We parameterize a value function $Q(s, a; \phi_l)$ using the deep neural network of MLPs in which $\phi_l$ are the parameters of the Q-network at updating iteration $l$. When reaching decision point $t_i$, we obtain the state $s_{t_i}$, action $a_{t_i}$, and reward $r_{t_i}$ according to Section 3.2.2 for the current static problem and save them to the replay buffer. When reaching decision point $t_{i+1}$, we obtain the state $s_{t_{i+1}}$, which is the next state of the previous static sub-problem and we get a new transition $e_{t_i} = (s_{t_i}, a_{t_i}, r_{t_i}, s_{t_{i+1}})$. We store the transitions into buffer $D = \{e_{t_1}, ..., e_{t_i}, ...\}$ during the running of simulator. During the training, we apply Q-learning updates on uniformly sampled transitions $(s, a, r, s') \sim U(D)$ from the replay buffer. The model updates at iteration $l$ uses the following Temporal Difference (TD) loss function:

$$L_i(\phi_l) = \mathbb{E}_{(s,a,r,s') \sim U(D)} \left[ \left( r + \gamma \max_{a'} Q(s', a'; \phi_l^-) - Q(s, a; \phi_l) \right)^2 \right] \tag{3}$$

where $\gamma$ is the discount factor, $\phi_l$ are the parameters of the Q-network at iteration $l$ and $\phi_l^-$ are the parameters of the target network at iteration $l$.

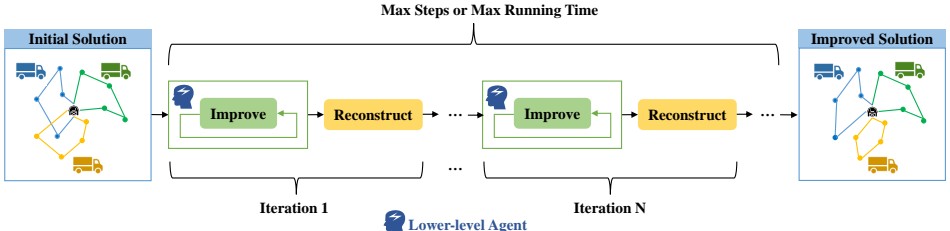

Figure 3: Workflow of Lower-level Agent

### 3.3 Lower-level Agent

#### 3.3.1 Workflow

The workflow of the lower-level agent is described in Figure 3. We first generate a feasible initial solution of the static sub-problem using greedy algorithm (described in Section 4.2). Given the initial solution, the lower-level agent iteratively improves the solution by manipulating different operators according to solution states as mentioned in Section 3.1 (we call this step *Improvement* [6, 13]). When the improved solution reaches a local optimum (i.e., the solution could not be improved further for a series of steps), we will partially or entirely re-assigning the orders using the greedy algorithm (we call this step *Reconstruction*). The improvement of the next iteration will start from the reconstructed solution. Note that the lower-level agent only selects improvement operators as reconstruction operator has a long-lasting effect on solutions compared with improvement operators and we found mixing up them will lead to instability during the training. The process of improvement and reconstruction alternates until reaching the maximum number of steps or the maximum allowable running time. The best generated solution during the improvements and reconstructions will be adopted to dispatch orders to vehicles. Note that the final accepted solution is ensured to be at least as good as the initial solution. Overall, by transferring the knowledge learned from previously solved PDPs to the new ones, the agent could efficiently and monotonically improve the solution quality. The MDP definition of the operators-manipulating procedure is described in the following subsection.

#### 3.3.2 Markov Decision Process (MDP)

**State:** The state of the current solution consists of the states of all nodes, i.e., $s = \{s_1, s_2, ..., s_{|V|}\}$, where $s_v$ is the state of node $v$. Each $s_v$ includes the position information, order information, vehicle information and objective-related information. Details can be found in Appendix F.

**Action:** The action set consists of 4 carefully designed operators, i.e., *inner-exchange*, *inner-relocate*, *inter-exchange* and *inter-relocate*. We provide a proof in Appendix G that **any feasible solution (including the optimal one) could be obtained by iteratively applying these 4 operators from any given initial solution**. Details are described in Appendix G.

**Reward:** We find that the total objective can be easily reduced by a large margin during the first few improvement steps of an initial solution or reconstructed solution in the experiments (See Figure 4). If we assign the actions at these improvement steps a large reward, it's not fair for the actions in the subsequent steps. This is because the actions in the subsequent steps also play important roles in improving solutions in the complex solution space. Therefore, the overall objective $\text{OBJ}_b$ of the sub-problem after the first iteration (e.g., Iteration 1 in Figure 3) is used as the baseline following [13]. For each subsequent iteration $i$, we first get the optimized objective after the iteration as $\text{OBJ}_i$ and then assign $(|\text{OBJ}_b - \text{OBJ}_i|)/n_i$ to all $n_i$ actions executed in iteration $i$ as reward.

#### 3.3.3 Agent Policy Network

The policy network of the lower-level agent inputs the state of the current solution and outputs action probabilities of length $|A|$ where $A$ is the set of operators. In our case, a critical challenge of designing the policy network is the number of orders and the number of available vehicles are different for each static sub-problem. As the quantity of the combination of orders and vehicles are extremely huge, we cannot train a separate model for every combination of different numbers of orders and vehicles. Thus, the desired model should be able to transfer the knowledge learned from the previously solved

problems and generalize to new problems of any scale without fine-tuning. Besides, the routes of a solution naturally form a certain topological graph structure as shown in Figure 3. Therefore, in this paper, we incorporate GIN (Graph Isomorphism Network)[30], a powerful Graph Neural Network (GNN), to be the basis of the policy network of the lower-level agent. We use the REINFORCE algorithm [28] to train the policy network. Details of the policy network are shown in Appendix H.

# 4 Offline Evaluation

## 4.1 Experiments Settings

We start with designing a simulator to shed light on the contributions of the proposed framework under more controlled settings. Details of the simulator can be found in Appendix C. To comprehensively verify the effectiveness of our approach on problems of different scales, we use four types of datasets of different sizes, i.e., 15 orders with 5 vehicles, 50 orders with 5 vehicles, 300 orders with 20 vehicles, 1000 orders with 50 vehicles (matching the practical problem of thousand scales). Note that the orders/vehicles ratios are set according to realistic business settings. Each type of datasets contains 10 datasets, including 7 training sets and 3 test sets according to the ratio of 7:3 (e.g., 300-1, 300-2 and 300-3 are test sets with 300 orders). The vehicles have the same load capacity. Details of the datasets are described in Appendix D. According to the realistic business settings, the time span between two consecutive decision points is set to 10 minutes in the simulator.

The comparisons of different methods proceed as follows. For our approach, we first train a shared model on each type of training datasets and then evaluate the model on the test datasets of the same size. At each decision point, the upper-level agent decides whether to release orders to the lower-level agent. The lower-level agent is executed for no more than 10 minutes after receiving orders from the upper-level agent. Both agents are trained simultaneously. This training process is relatively stable due to the following reasons. The iterative solution optimization process (starts from an initial greedy solution) of our lower-level agent can ensure the obtained solutions have relatively high quality even at the initial training stages. In other words, the solutions given by the lower-level agent are relatively stable. Therefore, the unstable issue of co-training both levels of policies in our case is negligible, and thus both levels can be trained simultaneously. For baselines showed in Section 4.2, we also run them for up to 10 minutes at each decision point. We run the simulator until all the orders of the dataset are dispatched and completed to ensure fair comparisons. All the results in the experiments are obtained by running each algorithm ten times to get the mean and variance value of the optimization objective.

## 4.2 Baselines

To help readers better understand the baselines, we name them in the format of 'upper-level method + lower-level method' except for the **Optimal** baseline. '10min-Interval' means the dynamic problem is partitioned into static sub-problems with a fixed interval of ten minutes. '1order-Interval' means the dynamic problem is partitioned into static sub-problems consists of a single order.

**10min-Interval + Greedy:** Greedy [15] is the most widely-used method in industry, which is also the online deployed baseline method. The solution routes are expanded by greedily inserting new pickup and delivery nodes until all the orders are inserted.

**10min-Interval + ALNS:** ALNS [29] is one of the most representative meta-heuristic local search frameworks for solving DPDP that uses a series of operators to improve the solution. In each iteration, an operator is selected to destroy the current solution, and an operator is selected to repair the solution.

**1order-Interval + E2ERL:** According to [14], we use a DQN model to assign vehicles to each generated order and insert each order into the vehicle's order queue using the Greedy algorithm. It's an E2ERL (end-to-end RL) algorithm.

**10min-Interval + ST-DDGN:** ST-DDGN [12] is the state-of-the-art method that predicts future orders of DPDP. Then both the predicted orders and real generated orders are considered when solving each sub-problem using E2ERL.

**Optimal:** We convert DPDPs to static PDPs as we can obtain all the orders' information beforehand in offline style. Then we use Gurobi to solve the corresponding MIP model to get the optimal solution. As Gurobi can only solve small-scale PDPs within acceptable time due to the NP-hard property, we only compare with the optimal solution on problems of 15 and 50 orders in Section 4.4.1.

## 4.3 Main Results

Here we show part of the experimental results in Table 1. The complete results are shown in Table 6 of Appendix I. The objective improvement measurement is the improvement percentage of each algorithm compared with the Greedy algorithm. **Our approach consistently outperforms all baselines**

Table 1: Main results of different methods on test datasets

| Dataset | Algorithm | Overtime | Avg_Dis | Objective | Obj Impro |
|---|---|---|---|---|---|
| | 10min-Interval + Greedy | 0 | 109.30 | 109.30 | 0.00% |
| | 1order-Interval + E2ERL | 0 | 96.56 | 96.56 | 11.66% |
| 50-1 | 10min-Interval + ALNS | 0 | 107.95 | 107.95 | 1.24% |
| | 10min-Interval + ST-DDGN | 0 | 108.95 | 108.95 | 0.32% |
| | **Our (Upper-level RL + Lower-level RL)** | **0** | **93.70** | **93.70** | **14.27%** |
| | 10min-Interval + Greedy | 0 | 147.78 | 147.78 | 0.00% |
| | 1order-Interval + E2ERL | 0 | 158.39 | 158.39 | -7.18% |
| 300-1 | 10min-Interval + ALNS | 0 | 137.31 | 137.31 | 7.08% |
| | 10min-Interval + ST-DDGN | 0 | 131.99 | 131.99 | 10.68% |
| | **Our (Upper-level RL + Lower-level RL)** | **0** | **122.42** | **122.42** | **17.16%** |
| | 10min-Interval + Greedy | 0 | 183.04 | 183.04 | 0.00% |
| | 1order-Interval + E2ERL | 0 | 180.36 | 180.36 | 1.46% |
| 1000-1 | 10min-Interval + ALNS | 0 | 174.68 | 174.68 | 4.57% |
| | 10min-Interval + ST-DDGN | 0 | 171.09 | 171.09 | 6.53% |
| | **Our (Upper-level RL + Lower-level RL)** | **0** | **159.18** | **159.18** | **13.04%** |

**on all datasets** (lower total objective is better). On some datasets, the baselines have overtime results due to their lack of long-term planning and exhaustively optimization of each static problem from a myopic perspective. As a result, dispatching of some orders is delayed for too long, and finally, overtime is inevitable in any case. In contrast to this, our upper-level RL partitions the dynamic problem into sub-problems considering the balance between the orders overtime (seconds) risk and optimization of vehicle travelling distances (kilometers), and our lower-level RL is responsible for the optimization of each static sub-problem. The cooperation of the two agents enables our method to find solutions with less overtime and vehicle travelling distances on the overall dynamic problem from a long-term perspective. The comparison of the learning curves of all learning-based methods on 50-1 are shown in Figure 11 in Appendix I.

## 4.4 Ablation Studies

### 4.4.1 How far is our lower-level agent from the optimal one on static PDP?

We convert the DPDP to a single static PDP as described in Section 4.2. As the orders should be assigned to vehicles all at once, there is no need to use an upper-level agent. Similarly, without the prediction of future orders, ST-DDGN is essentially the same as E2ERL. Therefore we only use the lower-level agent and E2ERL in the static PDP. Each algorithm is run without time or step limitation to discover its full potential. As shown in Table 2 and 3, the difference of the total objective of our method with the optimal solution is much smaller than the baselines (as all the overtime is 0, the column is omitted from the two Tables). The time consumption is much shorter than ALNS and Gurobi. It is because our method can exert the generalization ability to quickly improve the initial solution by using the most appropriate operators based on the experiences obtained from training, without the need of manually designing complicated search as in ALNS and Gurobi. Note that the time consumption of Gurobi on problems of 50 orders is represented using hyphen symbol '-', which means we can't get results even after 100 hours due to the various complex constraints as described in Appendix A. Comparing with baselines, our method is the most qualified to meet the online deployment requirements that the algorithm should obtain high-quality solutions with fast speed.

Table 2: Results on static 15-1, 15-2, 15-3

| Algorithm | Avg_Dis | Objective | Obj Impro | Time |
|---|---|---|---|---|
| Greedy | 53.85 | 53.85 | 0.00% | 0.38s |
| E2ERL | 51.70 | 51.70 | 3.99% | 0.58s |
| ALNS | 51.58 | 51.58 | 4.22% | 405s |
| **Our (Lower-level RL)** | **45.72** | **45.72** | **15.10%** | **68.21s** |
| Optimal | 44.35 | 44.35 | 17.64% | 141360s |
| Greedy | 69.61 | 69.61 | 0.00% | 0.40s |
| E2ERL | 68.00 | 68.00 | 2.31% | 0.79s |
| ALNS | 62.62 | 62.62 | 10.04% | 606s |
| **Our (Lower-level RL)** | **62.32** | **62.32** | **10.47%** | **27.96s** |
| Optimal | 57.48 | 57.48 | 17.43% | 193680s |
| Greedy | 78.73 | 78.73 | 0.00% | 0.34s |
| E2ERL | 59.02 | 59.02 | 25.03% | 0.83s |
| ALNS | 52.21 | 52.21 | 33.68% | 920s |
| **Our (Lower-level RL)** | **50.95** | **50.95** | **35.29%** | **71.98s** |
| Optimal | 50.75 | 50.75 | 35.54% | 28651s |

Table 3: Results on static 50-1, 50-2, 50-3

| Algorithm | Avg_Dis | Objective | Obj Impro | Time |
|---|---|---|---|---|
| Greedy | 98.68 | 98.68 | 0.00% | 66.73s |
| E2ERL | 94.94 | 94.94 | 3.79% | 52.64s |
| ALNS | 96.98 | 96.98 | 1.72% | 10728.34s |
| **Our (Lower-level RL)** | **82.43** | **82.43** | **16.47%** | **1459.23s** |
| Optimal | - | - | - | - |
| Greedy | 80.67 | 80.67 | 0.00% | 22.27s |
| E2ERL | 76.04 | 76.04 | 5.74% | 16.66s |
| ALNS | 65.31 | 65.31 | 19.04% | 6012.56s |
| **Our (Lower-level RL)** | **58.42** | **58.42** | **27.58%** | **1152.64s** |
| Optimal | - | - | - | - |
| Greedy | 83.34 | 83.34 | 0.00% | 44.31s |
| E2ERL | 80.92 | 80.92 | 2.90% | 20.34s |
| ALNS | 80.51 | 80.51 | 3.40% | 4140.15s |
| **Our (Lower-level RL)** | **72.66** | **72.66** | **12.81%** | **1998.09s** |
| Optimal | - | - | - | - |

### 4.4.2 Does lower-level agent learn how to select operators?

To verify that our lower-level agent learns to choose the most suitable operators at different states, we compare the results of different operator selection methods on both static problems and dynamic problems. We first compare the total objective during the solution improvement process on the static problems using the lower-level agent with the method that randomly selects operators. As the improvement-reconstruct iteration process designed in Section 3.3 ensures the quality of the solution can be monotonically improved, selecting operators randomly is also a powerful baseline that can achieve satisfactory performance for the static PDP.

Therefore, for the static problem, we mainly focus on whether the lower-level agent can improve the solving speed. As we can see in Figure 4, at the same step, choosing operators using lower-level RL can reach a better objective than randomly choosing operators, which indicates our lower-level method learned to accelerate the searching for better solutions. Since the static sub-problems of a dynamic problem are not independent of each other, the small gap between the above two methods in a static sub-problem will continue to enlarge in the subsequent sub-problems, resulting in a very large result gap on the entire dynamic problem. We compare the total objectives of the entire dynamic problems in Table 4. To ensure fairness, we control the upper-level methods to be '10min-Interval' and use the lower-level agent and random selection as lower-level methods, respectively. As we can see in Table 4, using the lower-level agent can help find better solutions on the entire dynamic problems.

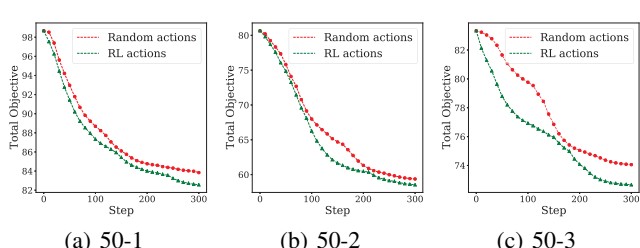

|           (a) 50-1           |           (b) 50-2           |           (c) 50-3           |

Figure 4: Comparison of different selection methods of operators on static problems

### 4.4.3 Does the upper-level agent learn to partition DPDP from a long-term perspective?

To verify that our upper-level agent learns to partition the dynamic problem into static sub-problems from a long-term perspective, we compare the results of differ-

Table 4: Effectiveness of lower-level agent and upper-level agent

| Dataset | Method | Overtime | Avg_Dis | Objective | Obj Impro |
|---------|--------|----------|---------|-----------|-----------|
| 300-1 | 10min-Interval + Random Search | 0 | $139.16 \pm 8.59$ | 139.16 | 0.00% |
|       | 10min-Interval + Lower-level RL | 0 | $126.95 \pm 4.80$ | 126.95 | 8.77% |
|       | **Our (Upper-level RL + Lower-level RL)** | **0** | **$122.42 \pm 4.02$** | **122.42** | **12.03%** |
| 300-2 | 10min-Interval + Random Search | 0 | $166.31 \pm 10.20$ | 166.31 | 0.00% |
|       | 10min-Interval + Lower-level RL | 0 | $154.39 \pm 8.13$ | 154.39 | 7.17% |
|       | **Our (Upper-level RL + Lower-level RL)** | **0** | **$142.33 \pm 7.47$** | **142.33** | **14.42%** |
| 300-3 | 10min-Interval + Random Search | 0 | $168.69 \pm 7.70$ | 168.69 | 0.00% |
|       | 10min-Interval + Lower-level RL | 0 | $156.64 \pm 7.50$ | 156.64 | 7.14% |
|       | **Our (Upper-level RL + Lower-level RL)** | **0** | **$146.88 \pm 13.91$** | **146.88** | **12.93%** |

ent static sub-problems partitioning methods. We compare our upper-level agent with the '10min-Interval' method. As we can see in Table 4, using an upper-level agent to partition the dynamic problem reaches the best objective. The results illustrate that our upper-level agent can partition the problem from a long-term perspective to balance the orders overtime risk and optimization of vehicle travelling distances.

### 4.4.4 Can our method generalized to larger-scale problems?

To verify our method's generalization ability, we evaluate the models trained using datasets of 300 orders / 20 vehicles on larger-scale datasets, i.e., 1000 orders / 50 vehicles. As shown in Table 5, the model trained on datasets of 300 orders achieves similar performance with the one trained on datasets of 1000 orders.

Table 5: Generalization verification

| Dataset | Model | Overtime | Avg_Dis | Objective |
|---------|-------|----------|---------|-----------|
| 1000-1 | Trained on 1000 | 0 | $159.18 \pm 4.10$ | 159.18 |
|        | Trained on 300 | 0 | $170.78 \pm 10.27$ | 170.78 |
| 1000-2 | Trained on 1000 | 0 | $196.66 \pm 9.52$ | 196.66 |
|        | Trained on 300 | 0 | $209.48 \pm 8.68$ | 209.48 |
| 1000-3 | Trained on 1000 | 0 | $176.39 \pm 7.61$ | 176.39 |
|        | Trained on 300 | 0 | $182.83 \pm 5.64$ | 182.83 |

Note that the model trained on datasets of 300 orders also outperforms the baselines in Table 1. It verifies that our method can be generalized to new problems of different scales without fine-tuning after well trained on existing problems.

## 5 Online Testing

We deployed our method on
the order dispatching system
in Huawei supply chain. In the
online experiments, we com-
pare our method with the pre-
viously online deployed greedy
algorithm (10min-Interval +
Greedy). For a fair compar-
ison, we control the vehicles
and the nodes (factories and
warehouses) involved in the
online testing to be the same.
Normally, a standard A/B test-
ing is required to be performed
on homogeneous experimental
groups using different methods

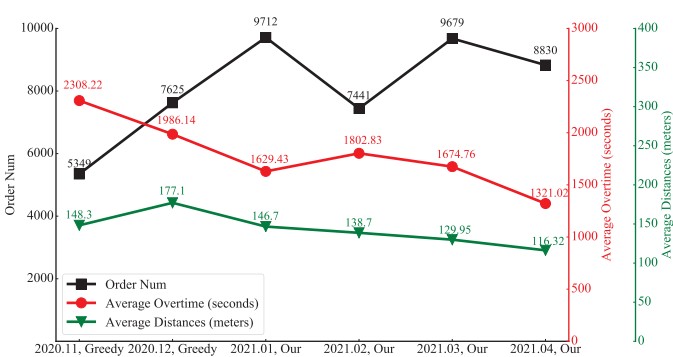

Figure 5: Online Results

at the same time dimension. Then experimental data of each group are collected and evaluated to
choose the best method. However, in our situation, it's unrealistic to split each order into two
sub-orders to ensure the experimental groups are homogeneous. Besides, a large number of offline
experiments have demonstrated that our method is significantly better than the greedy algorithm.
Even in the worst case when there is no improvement in each sub-problem, our method is still the
same with Greedy. Thus, we directly replaced the greedy algorithm for online deployment. Figure 5
summarises the results from Nov 2020 to Apr 2021. The points of Nov and Dec 2020 shown in Figure
5 are generated by greedy algorithm and our method is deployed from Jan to Apr 2021. As we can
see, our method can reduce the average orders' overtime and vehicles' travelling distances compared
with the greedy baseline. Even with more orders, our method can still reach a better optimization
objective. Note that in the actual business scenario, orders generated in each day follow a similar
distribution with a small variance. These results indicate that our method could achieve a better
performance in the realistic deployment environment with varied data distributions.

## 6 Conclusions

In this paper, we propose a novel hierarchical reinforcement learning based optimization framework
to solve the large-scale DPDP in the real world. The upper-level agent is equipped with the far-sight
ability whose target is to optimize the long-term cumulative objective. The lower-level agent exerts the
generalization ability of GNN to quickly improve the solution quality by transferring the knowledge
(policy) learned from training. The cooperation of the upper-level and lower-level agents enables
our method to find globally better solutions. Extensive offline simulation on the simulator built on
real historical data and online testing verify that our method can obtain higher-quality solutions with
faster running speed.

The core idea of our learning-based framework are beneficial to a number of similar problems in the
supply chain community that have time-evolving components (e.g., orders/customers/tasks), such as
dynamic routing problems, dynamic flow shop scheduling, dynamic job shop scheduling, dynamic
bin packing and so on. As orders/customers/tasks of all these dynamic problems are online generated
that are not known a priori, the orders/customers/tasks should first be cached and then be dispatched.
In this way, these problems can be modeled as hierarchical optimization problems like DPDP that the
upper-level problem is "how to cache orders/customers/tasks" and the lower-level problem is "how to
dispatch cached orders/customers/tasks". We will verify our proposed framework in these fields in
the future work.

## Acknowledgments and Disclosure of Funding

The work is supported by the National Natural Science Foundation of China (Grant Nos: U1836214)
and the New Generation of Artificial Intelligence Science and Technology Major Project of Tianjin
under grant: 19ZXZNGX00010.

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
