# A PDP Formulation

A static Pickup and Delivery Problem (PDP) can be formulated as an MIP as follows:

$$\min \frac{1}{K} \sum_{k \in K} \sum_{i \in V} \sum_{j \in V} d_{ij} \times x_{ij}^k + \lambda \sum_{k \in K} \sum_{i \in P} \max\{0, T_{n+i}^k - l_i\} \tag{4}$$

$$subject\ to \sum_{k \in K} \sum_{j \in P \cup D, j \neq i} x_{ij}^k = 1, \ \forall i \in P, \tag{5}$$

$$\sum_{j \in V} x_{i,j}^k - \sum_{j \in V} x_{j,n+i}^k = 0, \ \forall i \in P, \ \forall k \in K, \tag{6}$$

$$\sum_{j \in V} x_{j,i}^k - \sum_{j \in V} x_{i,j}^k = 0, \ \forall i \in P \cup D, \ \forall k \in K, \tag{7}$$

$$\sum_{w \in W} \sum_{i \in P \cup W} x_{w,i}^k = 1, \ \forall k \in K, \tag{8}$$

$$\sum_{w \in W} \sum_{i \in D \cup W} x_{i,w}^k = 1, \ \forall k \in K, \tag{9}$$

$$Q_j^k \geq (Q_i^k + q_j) x_{ij}^k, \ \forall i, j \in V, \ \forall k \in K, \tag{10}$$

$$0 \leq Q_i^k \leq Q, \ \forall i \in V, \ \forall k \in K, \tag{11}$$

$$Q_i^k = 0, \ \forall i \in W, \ \forall k \in K, \tag{12}$$

$$Q_{n+i}^k = Q_i^k - q_i, \ \forall i \in P, \ \forall k \in K, \tag{13}$$

$$T_j^k \geq (T_i^k + t_{ij} + s_i) x_{ij}^k, \ \forall i, j \in V, \ \forall k \in K, \tag{14}$$

$$T_i^k \geq e_i, \ \forall i \in V, \ \forall k \in K, \tag{15}$$

$$T_{n+i}^k \geq T_i^k + t_{i,n+i} + s_i, \ \forall i \in P, \ \forall k \in K, \tag{16}$$

$$x_{ij}^k \in \{0, 1\}, \ \forall i, j \in V, \ \forall k \in K, \tag{17}$$

$$T_i^k \geq 0, \ \forall i \in V, \ \forall k \in K. \tag{18}$$

where

- $n$: the number of orders in the static PDP
- $W = \{d_1, d_2, ..., d_w\}$: the set of depots, recording initial positions of vehicles
- $C = \{c_1, c_2, ..., c_n\}$: the set of orders[5] in the static PDP, $c_i$ contains pickup node $i$ and delivery node $n + i$
- $P = \{1, 2, ..., n\}$ is the set of pickup nodes and $D = \{n + 1, n + 2, ..., 2n\}$ is the set of delivery nodes in the static PDP
- $V = W \cup P \cup D$: the set of all nodes in the static PDP
- $K_w = \{k_{w,1}, k_{w,2}, ..., k_{w,k_w}\}$: the set of vehicles in depot $w$
- $k_w$: the number of vehicles in depot $w$
- $K = \sum_{w \in W} K_w$: the set of total vehicles
- $Q$: maximum loading capacity of each vehicle
- $q_i$: demand of order $i$ (positive for pickup and negative for delivery operations)
- $Q_i^k$: the load of vehicle $k$ after leaving node $i$
- $T_i^k$: the time when vehicle $k$ begins to serve node $i$
- $[e_i, l_i]$: the delivery time window of request $i$
- $d_{ij}$ and $t_{ij}$ is the distance and the traveling time between node $i$ and $j$, respectively
- $x_{ij}^k$: a binary decision variable, and $x_{ij}^k = 1$ if node $i$ is visited before node $j$ by vehicle $k$

The optimal solution of this MIP could be obtained utilizing exact algorithms (e.g., cutting plane algorithms, branch-and-bound algorithms or modern solvers such as Gurobi[17]). During the solution computing of above MIP, following constraints should be satisfied:

---

[5]Some orders contain large amount of goods that cannot be loaded onto one vehicle, so we split them into multiple small orders, each of which contains one pallet of goods.

- Pickup and Delivery Constraint: Each good should be picked up first and then be transported to the destination by the same vehicle.

- Capacity Constraint: The capacity of vehicle cannot be exceeded [25].

- LIFO Constraint: Among all goods on board a vehicle, we must deliver the good that is most recently picked up to minimize the loading and unloading times [4, 5, 2, 21].

- Time Window Constraint: The completion time of each order should not exceed the committed maximum service time. As Time Window Constraint has been converted to a soft one, it can be violated while bringing punishment in the objective function.

- Split Orders Constraint: If an order's demand exceeds the vehicle's maximum capacity, the the order should be split into smaller orders, each of which contains goods with the smallest unit of measure (e.g., one pallet) and served by different vehicles [19, 10].

Below are the detailed constraints reflected in the MIP model. Constraint (5) ensures that each pickup location is visited. Constraint (6) ensures that the delivery location is visited if the pickup location is visited and that the visit is performed by the same vehicle. Constraint (7) ensures that a consecutive route is formed for each vehicle. Constraint (8) and (9) ensure that each vehicle starts from one depot and enters one depot. Constraints (10) $\sim$ (12) ensure that the load of vehicles is set correctly and capacity constraints are satisfied. Constraint (13) ensures that the last-in-first-out (LIFO) rule is obeyed. Constraint (14) and (15) ensure that $T_j^k$ is set correctly on the route. Constraint (16) ensures that each pickup occurs before the corresponding delivery.

# B  Literature Reviews

**Traditional methods for DPDP.** Existing solutions for DPDP fall into two classes. The first class of methods partitions the entire dynamic problem into a series of static sub-problems, each of which only contains a small subset of recently cached orders. Then, the dynamic problem is solved by sequentially compute the solution of each sub-problem using operational research-based (OR) methods [22], heuristic methods or meta-heuristic methods (use a meta controller to schedule heuristic operators, e.g., change two nodes, to improve a given initial solution iteratively) [7, 16, 3, 11, 26, 23, 9]. However, myopically optimizing each static sub-problem doesn't guarantee the overall dynamic problem will be optimized as the static sub-problems are not independent of each other. The main reasons are previous orders assignment results will influence (1) the number of remaining orders to be dispatched, (2) the vehicle's remaining capacity and (3) the relative positions to the following orders. Therefore, considering this fact, the second class of methods predicts the distribution of future orders and take the predicted orders into consideration when computing the solution for each static sub-problem. Thus the vehicles will have more candidate orders (including the real generated orders and the predicted orders) to choose. For example, [24] builds a random generative model and [8] samples orders from historical logs to simulate the generation of future orders. However, for the real-world problems, it is hard to predict future orders correctly due to the high uncertainty. The uncertainty derives from several perspectives such as market fluctuations, special events (promotion days or holidays), orders cancellation, traffic and weather conditions, etc.

In addition to the limitations mentioned above, these traditional methods have some flaws that make them difficult to be applied to real-world large-scale DPDPs: (1) The design of them heavily relies on complex domain knowledge; (2) Due to the high computational complexity of DPDP, these methods are time-consuming due to their exhaustive searching process.

**Learning-based methods for VRP.** Traditional algorithms mentioned above could be regarded as a series of decision-making processes that are designed manually according to complex domain knowledge. With the development of deep neural networks (DNNs) and reinforcement learning (RL), it has been proved that DNN-parameterized RL policies can be used to make decisions in various scenarios, e.g., playing Atari games [17]. Therefore, to improve the defects of the traditional algorithms, more and more learning-based methods are designed to solve vehicle routing problems in recent years. These methods do not require manual design of complex rules and can achieve competitive results with faster solving speed compared with the traditional approaches. Most learning-based methods for routing problems focus on TSP (visiting each node, i.e., city, with shortest total travelling distances) [27, 1] and VRP (arranging vehicles to serve nodes, i.e., customers, with least total route cost) [18, 6, 13]. These methods can be generally divided into two classes. The first class of methods follows an end-to-end design, which directly constructs a solution from scratch [27, 1, 18]. [27] first introduces the Pointer Network (PtrNet) to solve TSP, which incorporates an Attention model to learn the visiting order of different nodes by supervised learning. Based on [27], [1] further improve the solution quality of the PtrNet using RL algorithms. [18] improves the PtrNet by designing a model structure invariant to the nodes' input sequence. However, these methods have unsatisfactory performances on large-scale problems. The second class of methods [6, 13] adopts the ideas similar to the meta-heuristic algorithms, which models the operators' scheduling process as a sequential decision process to speed up the searching for good solutions. [6] divides the nodes into several regions and trains a region-picking policy to select which region to improve at each state. An operator-picking policy is also trained to choose operators to improve the selected region. [13] uses an RL policy

to decide which operators should be used to improve the current solution at each state and a rule to reconstruct the current solution when reaching local-optimal. However, these methods focus only on static VRPs, of which all orders' information is known in advance and much fewer constraints are considered comparing with the DPDP.

## C  Simulators

The simulator is used to simulate the generation of orders within a period of time, the assignment of orders to vehicles, the transportation of goods by vehicles, the updating of vehicle locations, etc.

### C.1  Structure of the Simulator

The main components of the simulator are: (1) Scheduling module is the entrance of the simulator and decides the selection of algorithms; (2) Data-process module is to prepare the data used in the simulator; (3) Simulation module is the main module for simulating the behaviors of the vehicles and the orders; (4) Statistics module is used to calculate various indicators (e.g., vehicles travelling distances, algorithm running time) for the solved problems.

### C.2  Design of the Simulator

#### C.2.1  Distance Matrix

To simulate the realistic transportation network, an actual travel distance matrix based on real industrial historical data is obtained instead of the Euclidean distance matrix to construct the simulator. In this actual travel distance matrix, the distances between the same two nodes may be different due to different route directions. For example, for node A and B, the distances of route A->B and route B->A may be different.

#### C.2.2  Time dictionaries

The simulation is based on two time dictionaries: time_dict_arrive and time_dict_depart, in both of which vehicle ID is the key, time list is the value. Time list for each vehicle in time_dict_arrive records the arrival time of each vehicle to the nodes on the scheduled routes, and that in time_dict_depart is the departure time to the nodes.

For the scheduled routes, the time of arrival and departure to each node is computed in an accumulative way. Taking an ordinary scheduled route for example, a route is composed of a depot and several ordered nodes, in which the depot is the vehicle's initial position in each static sub-problem while a node can be either for pickup or delivery. Given the start time, which is the last updated time, the latitude and longitude of the vehicles, depot, and the nodes, the distance of every single route can be computed using the real distances in the actual travel distance matrix. If the position of the vehicle is not a depot or a node, that is, the true distance between the vehicle's current position and its next destination cannot be obtained in the distance matrix, then we approximate the real distance through the following formula:

$$S = 2 \times 6371.137 \times \tag{19}$$

$$\arcsin \sqrt{\sin^2(a/2) + \cos(lat1) \times \cos(lat2) \times \sin^2(b/2)} \tag{20}$$

in which $a = lat1 - lat2$, $b = lon1 - lon2$, and $6371.137km$ is the radius of the earth. $(lat1, lon1)$ and $(lat2, lon2)$ denote the latitude-longitude pair of the beginning and the end of every single route. To compute the corresponding time consumption, the vehicle speed is set to 3 kilometers per minute for the vehicle traveling in the company's industrial park while 1.5 kilometers per minute outside the industrial park. The speed setting is an average value according to the statistics of real historical data. Given the pickup and delivery time needed for loading or unloading the goods, the time consumed at the depot or nodes can be computed. Hence, we can compute the estimated arrival and departure time to the depot or nodes on the scheduled route.

#### C.2.3  Simulating Vehicles' Position

Based on the two time dictionaries described above, vehicles' positions can be simulated. Since the start time of the time dictionaries is the last updated time, the time interval between the last updated time and the current time can be computed, for instance, 10 minutes. For each vehicle, we can utilize this time interval and time list for the vehicle to locate it. As shown in Figure 6, 10 minutes after the last updated time, the vehicle left node 1 and is on the way to node 2. To compute the exact location of the vehicle, we utilize the following formula, which computes the latitude and longitude of an intermediate point at any fraction along the great circle path between

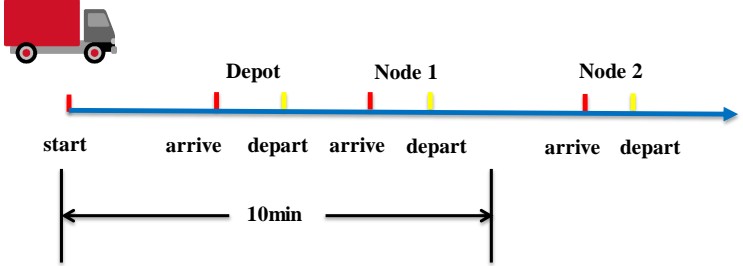

Figure 6: Simulating Vehicle's Position

two points:

$$a = \sin((1 - f) \cdot \delta) / \sin \delta, \tag{21}$$
$$b = \sin(f \cdot \delta) / \sin \delta, \tag{22}$$
$$x = a \cdot \cos \varphi_1 \cdot \cos \lambda_1 + b \cdot \cos \varphi_2 \cdot \cos \lambda_2, \tag{23}$$
$$y = a \cdot \cos \varphi_1 \cdot \sin \lambda_1 + b \cdot \cos \varphi_2 \cdot \sin \lambda_2, \tag{24}$$
$$z = a \cdot \sin \varphi_1 + b \cdot \sin \varphi_2, \tag{25}$$
$$\varphi_i = a \tan 2(z, \sqrt{x^2 + y^2}), \tag{26}$$
$$\lambda_i = a \tan 2(y, x), \tag{27}$$

in which $f$ is fraction along great circle path ($f = 0$ is node1, $f = 1$ is node2), $\delta$ is the angular distance $d/R$ between the two points[6]. In Figure 6, as the latitude and longitude of node 1 and node 2, and fraction of distance traveled by vehicle between node 1 and node 2 are known, we can compute the current location of the vehicle.

### C.2.4   Simulating Orders' Status

In the same way, we can simulate the orders' status. There are 4 types of status for each order: (1) not picked up; (2) being picked up; (3) on the vehicle; (4) delivered. Comparing the time elapses and the time list of the vehicle, we have the following cases:(1) Case 1, Depot/Node is not visited yet: For depot or a delivery node, the order to be delivered at the depot or the delivery node is still in status 3, i.e., on the vehicle; for a pickup node, order to be picked up is in status 1; (2) Case 2, Depot/Node is being visited: For depot or a delivery node, the order is being delivered, is still labeled status 3, i.e., on the vehicle; for a pickup node, the order being picked up is labeled status 2. (3) Case 3, Depot/Node has been visited: For depot or a delivery node, the order is delivered, hence, the status is changed to be 4; for a pickup node, the order is picked up, hence, the status is changed to 3.

## D   Datasets

The datasets used in the simulator are extracted from the real historical data generated in 102 factories and warehouses of Huawei supply chain order dispatching system, ranged from Sep 2019 to Mar 2020, in which information of the requests includes order ID, demand, pickup node, delivery node, creation time, and committed completion time, etc. To obtain the data of a single day, the date information of the orders is dismissed. To keep the original pattern of the data, we intergrate all the historical data samples and choose the data randomly with equal probability weighting. The dataset consists of DPDPs of a varying number of orders, that is, $\{15, 50, 300, and 1000\}$. For each number of orders, ten problems are generated.

## E   Details of Implementation

Experiments were performed in PyTorch. The hardware we use is a server group of Ubuntu 18.04.4LTS with 111 Intel(R) Xeon(R) Platinum 8180M CPUs@2.50GHz and 2 Nvidia Tesla V100 GPU. For the upper-level agent, the DQN model is a two-layer MLP with one ReLU as the activation function. The epsilon value is set to 0.9 and the discount factor is set to 0.99. For the lower-level agent, the basis of the policy network is a GIN. The number of GIN embedding layers is set to 2. Too many embedding layers will decrease the performance of the model because using more layers would make the model easier to lose low-layer information. The pooling method used sum pooling. The dropout rate of the final layer is set to 0.5. Then the output of the GIN is sent to an MLP and a softmax layer to calculate the probability of each action. All the optimizers used in our experiments are Adam optimizers with learning rates 0.01 and 0.001 for DQN and REINFORCE, respectively.

---

[6]https://www.movable-type.co.uk/scripts/latlong.html

# F  Details of states

For the upper-level agent, the states include:

- The total number of orders accumulated to the present since the last releasing of orders;
- The total available vehicle number at present;
- The reward information obtained in the last finished static sub-problem, including the total overtime and vehicle travelling distances of the orders finished in the static sub-problem;
- The time remaining before the committed delivery time of the earliest cached order. For example, if the committed delivery time of the earliest cached order is 10 am while currently it's 8 am, then the remaining time is 2 hours. As the scale of time is too large compared with other features, the remaining time is normalized by dividing committed maximum service time (e.g., 288000 seconds).

For the lower-level agent, the states are the concatenation of all the nodes features in all the current routes. For each node $i$, the features include:

- The position information $(x_i, y_i)$ of node $i$ where $x_i$ represents latitude of $i$ and $y_i$ represents longitude of $i$;
- Demand of node, positive value indicates the vehicle should load goods at node $i$ and negative value indicates the vehicle should unload goods at node $i$ ;
- Vehicle's left capacity after loading or unloading goods at node $i$;
- Vehicle's travelling distance when arriving at node $i$;
- Total distance of the route that node $i$ belongs to;
- Total overtime of the route that node $i$ belongs to;

# G  Design of operators (actions)

Now we introduce the detailed design of operators. In Figure 7, 8 and 9, different colors indicates different vehicles. $P_i$ and $D_i$ represents the pickup node and delivery node of order $i$, respectively. We use two types of operators as the actions of the lower-level agent: inner-route operators to adjust nodes in one route of the solution and inter-route operators to adjust nodes in multiple routes of the solution.

- Inner-route:
    - Inner-Exchange: Swap two nodes inner the same route.
    - Inner-Relocate: Relocate an order's pickup node and delivery node to new locations in the current route.
- Inter-route:
    - Inter-Exchange: Swap pickup nodes and delivery nodes for two orders that belong to different routes.
    - Inter-Relocate: Relocate an order's pickup node and delivery node to new locations in another route.

We conclude that *by combining these operators to improve any initial solutions, each desired solution in the solution space can be obtained.* An illustrated proof is showed in Figure 7 in Appendix G. The grey lines and arcs indicate the operators applied on the solution. As we can see, the entire solution space can be traversed by iteratively applying the four operators on solutions. Note that all the constraints in Section 2 must be met when using operators.

In our initial experiments, we found that if the randomness contained in the operators was too high, the training of lower-level RL would be unstable. Therefore, we design simple rules for each operator to help stabilize the training as described below.

For the Inner-Exchange operator showed in Figure 8(a), we choose a route randomly and then choose an anchor order in the route randomly. Then we search for a pair of pickup and delivery nodes to change with the nodes of anchor order greedily so that the objective of the route after exchanging is minimized. For the Inner-Relocate operator showed in Figure 8(b), we choose a route randomly and then choose an anchor order in the route randomly. Then we search for a position in the route to insert the nodes of anchor order greedily so that the objective of the route after relocating is minimized.

For inter-route operators, we first describe the definition of the distance between two routes using the longitude $g_i$ and latitude $h_i$ information of each node $i \in O_r$ in the route, where $N_r$ is the set of nodes in route $r$. Given a

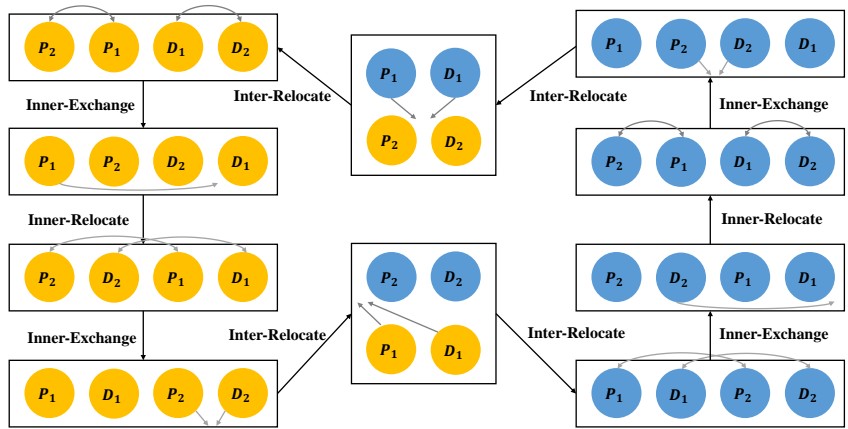

Figure 7: Coverage of solution space

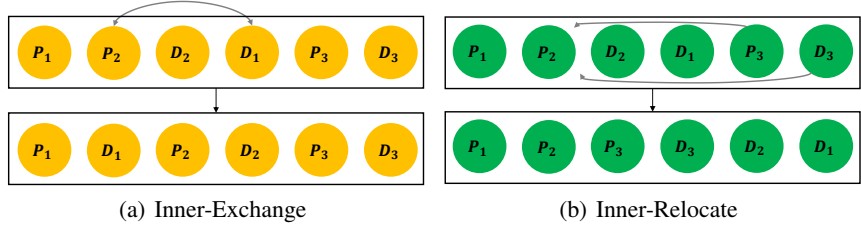

(a) Inner-Exchange      (b) Inner-Relocate

Figure 8: Inner-Route Operators

route $r$, we first calculate the average longitude $g_r = \frac{\sum_{i \in N_r} g_i}{|N_r|}$ and average latitude $h_r = \frac{\sum_{i \in N_r} h_i}{|N_r|}$ of each node in $r$. Then the distance between arbitrary two routes $r_1$ and $r_2$ can be calculated by $route\_dist(r_1, r_2) = |g_{r_1} - g_{r_2}| + |h_{r_1} - h_{r_2}|$. For the Inter-Exchange operator showed in Figure 9(a), we randomly choose one of the longest routes as the anchor route. Then we select a feasible route from all other routes in a near-to-far manner according to $route\_dist$ for node exchanging. The two orders to be exchanged are selected randomly, one from a route. The pickup nodes and delivery nodes of two orders are exchanged, respectively. The two routes after node exchanging with minimal total objectives will be returned. For the Inter-Relocate operator showed in Figure 9(b), we randomly choose one of the shortest routes as the anchor route, then we select a feasible route from all other routes in a near-to-far manner according to $route\_dist$ for node relocating. An anchor order in the anchor route is selected randomly. The pickup node and delivery node of the anchor order are relocated to the selected route from the neighborhood in a greedy way. The two routes after node relocating with minimal total objectives will be returned.

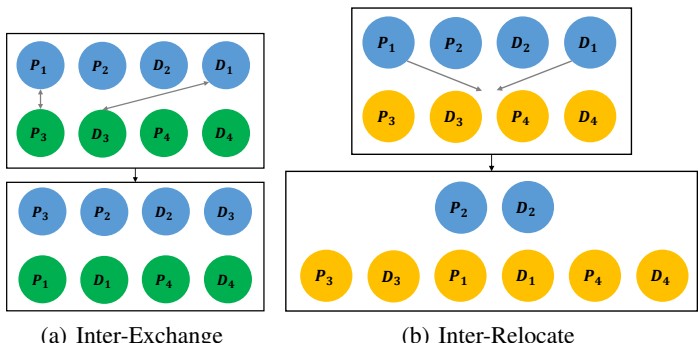

(a) Inter-Exchange      (b) Inter-Relocate

Figure 9: Inter-Route Operators

## H  Details of lower-level agent's policy network

Here we introduce the GNN-based policy network of the lower-level agent. The solution can be represented as a graph $G$. The depots of vehicles, the pickup nodes and delivery nodes of all orders together form the nodes in the graph. For node $v$, the contained features $s_v, v \in V$ are described in Section 3.3.2. The routes of each vehicle form the edges between nodes. For a node $v$, the policy network uses the neighborhood aggregation scheme in each layer to recursively aggregate the representation of adjacent nodes to calculate the representation vector of the node. For the given graph $G$, The $k$-th layer of the policy network is represented as follows:

$$h_v^{(k)} = MLP^{(k)} \left( \left( 1 + \varepsilon^{(k)} \right) \cdot h_v^{(k-1)} + \sum_{u \in N(v)} h_u^{(k-1)} \right) \tag{28}$$

where $h_v^{(k)}$ is the feature of node $v$ in $k$-th layer, $N(v)$ is the set of neighbor nodes of node $v$, $\varepsilon$ is a weight hyper-parameter, and MLP is a multi-layer perceptron. $h_v^{(0)}$ is initialized as $s_v$. The last layer of GIN uses a READOUT function (e.g., sum function) to aggregate the node features and concatenates ($CONCAT$) the embedding from all layers of the models to retain all structural information. Then the produced embedding of the entire graph is:

$$h_G = \text{CONCAT} \left( READOUT \left( \left\{ h_v^{(K)} \mid v \in G \right\} \right) \mid k = 0, 1, \ldots \right) \tag{29}$$

After obtaining the graph embedding vector, we use an MLP to map the graph embedding vector to the action space and use the softmax layer to obtain the action probability vector. Then the lower-level agent samples the actions to improve the solution according to the action probability distribution:

$$p_\theta(\pi \mid s) = \text{SOFTMAX} \left( \text{MLP} \left( h_G \right) \right) \tag{30}$$

where $\theta$ is the parameters of the policy network, $s = \{s_1, s_2, ..., s_{|V|}\}$ is the state of current solution and $\pi$ represents the current policy.

We use the REINFORCE algorithm [28] to update the gradient of the policy network:

$$\nabla_\theta J(\theta \mid s) = \mathbb{E}_{\pi \sim p_\theta(.|s)} \left[ (\text{OBJ}(\pi \mid s) - \text{OBJ}_b(s)) \nabla_\theta \log p_\theta(\pi \mid s) \right] \tag{31}$$

where $\text{OBJ}(\pi \mid s)$ is the total objective of solution obtained using $\pi$, and $\text{OBJ}_b(s)$ is the baseline as we described in Section 3.3.2.

Overall architecture of the lower-level agent's policy network is shown in Figure 10.

Figure 10: Architecture of Lower-level Agent's Policy Network

## I  Supplemental Experimental Results

For each method, we trained a model for each type of training datasets. Each model is trained for ten epochs on each dataset. After each epoch's training, we evaluated the model on the same size testing datasets for ten epochs to get Figure 11. As we can see, our method and E2ERL both gradually converge as the training goes.

We performed experiments under various hyperparameters and verified our method is not sensitive to hyperparameters. The hyperparameters of our method are mainly constituted of two classes: 1) the hyperparameters of GIN architectures; 2) the hyperparameter settings of RL algorithms. For each class of hyperparameters, a number of candidate hyperparameters are generated based on the settings of GIN [30], DQN [17] and REINFORCE [28]. Then, we got the optimal hyperparameters via grid search from these candidate hyperparameters and presented the corresponding results in our paper. During grid search, we found our method is robust under hyperparameter variations.

Table 6: Complete results of different methods on test datasets

| Dataset | Algorithm | Total Overtime | Average Distances | Total Objective | Obj Impro |
|---|---|---|---|---|---|
| 50-1 | 10min-Interval + Greedy | 0 | 109.30 | 109.30 | 0.00% |
| | 1order-Interval + E2ERL | 0 | 96.56 | 96.56 | 11.66% |
| | 10min-Interval + ALNS | 0 | 107.95 | 107.95 | 1.24% |
| | 10min-Interval + ST-DDGN | 0 | 108.95 | 108.95 | 0.32% |
| | **Our (Upper-level RL + Lower-level RL)** | **0** | **93.70** | **93.70** | **14.27%** |
| 50-2 | 10min-Interval + Greedy | 0 | 107.83 | 107.83 | 0.00% |
| | 1order-Interval + E2ERL | 0 | 104.56 | 104.56 | 3.03% |
| | 10min-Interval + ALNS | 0 | 106.93 | 106.93 | 0.83% |
| | 10min-Interval + ST-DDGN | 0 | 96.97 | 96.97 | 10.07% |
| | **Our (Upper-level RL + Lower-level RL)** | **0** | **88.82** | **88.82** | **17.63%** |
| 50-3 | 10min-Interval + Greedy | 0 | 115.39 | 115.39 | 0.00% |
| | 1order-Interval + E2ERL | 0 | 117.59 | 117.59 | -1.91% |
| | 10min-Interval + ALNS | 0 | 113.21 | 113.21 | 1.89% |
| | 10min-Interval + ST-DDGN | 0 | 114.12 | 114.12 | 1.10% |
| | **Our (Upper-level RL + Lower-level RL)** | **0** | **105.78** | **105.78** | **8.33%** |
| 300-1 | 10min-Interval + Greedy | 0 | 147.78 | 147.78 | 0.00% |
| | 1order-Interval + E2ERL | 0 | 158.39 | 158.39 | -7.18% |
| | 10min-Interval + ALNS | 0 | 137.31 | 137.31 | 7.08% |
| | 10min-Interval + ST-DDGN | 0 | 131.99 | 131.99 | 10.68% |
| | **Our** | **0** | **122.42** | **122.42** | **17.16%** |
| 300-2 | 10min-Interval + Greedy | 0 | 169.79 | 169.79 | 0.00% |
| | 1order-Interval + E2ERL | 0 | 167.13 | 167.13 | 1,57% |
| | 10min-Interval + ALNS | 0 | 164.65 | 164.65 | 3.03% |
| | 10min-Interval + ST-DDGN | 0 | 157.51 | 157.51 | 7.23% |
| | **Our (Upper-level RL + Lower-level RL)** | **0** | **142.33** | **142.33** | **16.17%** |
| 300-3 | 10min-Interval + Greedy | 0 | 188.30 | 188.30 | 0.00% |
| | 1order-Interval + E2ERL | 0 | 178.53 | 178.53 | 5.19% |
| | 10min-Interval + ALNS | 0 | 172.38 | 172.38 | 8.45% |
| | 10min-Interval + ST-DDGN | 0 | 171.50 | 171.50 | 8.92% |
| | **Our (Upper-level RL + Lower-level RL)** | **0** | **146.88** | **146.88** | **22.00%** |
| 1000-1 | 10min-Interval + Greedy | 0 | 183.04 | 183.04 | 0.00% |
| | 1order-Interval + E2ERL | 0 | 180.36 | 180.36 | 1.46% |
| | 10min-Interval + ALNS | 0 | 174.68 | 174.68 | 4.57% |
| | 10min-Interval + ST-DDGN | 0 | 171.09 | 171.09 | 6.53% |
| | **Our (Upper-level RL + Lower-level RL)** | **0** | **159.18** | **159.18** | **13.04%** |
| 1000-2 | 10min-Interval + Greedy | 5950 | 213.02 | 59,500,213.02 | 0.00% |
| | 1order-Interval + E2ERL | 3457 | 235.98 | 34,570,235.98 | 41.90% |
| | 10min-Interval + ALNS | 6945 | 187.34 | 69,450,187.34 | -16.72% |
| | 10min-Interval + ST-DDGN | 786 | 203.19 | 7,860,203.19 | 86.79% |
| | **Our (Upper-level RL + Lower-level RL)** | **0** | **196.66** | **196.66** | **≫ 100.00%** |
| 1000-3 | 10min-Interval + Greedy | 0 | 192.28 | 192.28 | 0.00% |
| | 1order-Interval + E2ERL | 0 | 188.45 | 188.45 | 2.00% |
| | 10min-Interval + ALNS | 2938 | 189.27 | 29,380,189.27 | ≪ -100.00% |
| | 10min-Interval + ST-DDGN | 0 | 185.64 | 185.64 | 3.45% |
| | **Our (Upper-level RL + Lower-level RL)** | **0** | **176.39** | **176.39** | **8.26%** |

Table 7: Comparison of different time span between two consecutive decision points on 300-1 using greedy algorithm

| 300 Orders / 20 Vehicles | 10 mins | 20 mins | 30mins |
|---|---|---|---|
| Total Overtime | 0 | 0 | 0 |
| Average Distances | 147.78 | 101.70 | 87.31 |
| Total Objective | 147.78 | 101.70 | 87.31 |

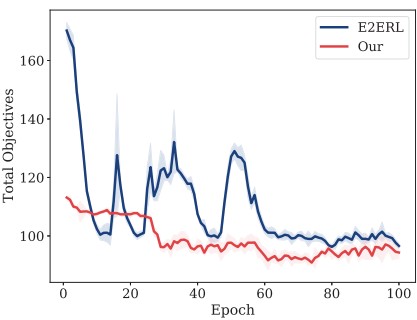

Figure 11: Average Evaluation Results on 50-1