# OpenReview forum: "A Hierarchical Reinforcement Learning Based Optimization Framework for Large-scale Dynamic Pickup and Delivery Problems"
_NeurIPS.cc/2021/Conference — NeurIPS 2021 Poster_

### Official Review · Reviewer_8RBm · 2021-07-16

**Rating:** 6
**Confidence:** 4

**Summary:**

The authors propose an approach to solve the Dynamic Pickup and Delivery Problem. The main challenge of this problem is that the orders are not known a priori. To solve this, the authors propose a bi-level approach composed of: (1) an upper-level agent to decide whether to solve the problem for current orders in a cache or to wait for new orders, and (2) a lower-level agent based on a Graph Isomorphism Network to solve the problem. Both agents are trained using classical reinforcement learning techniques. According to the results, this approach improves state of the art by showing superior performance over existing baselines for DPDP.

**Limitations And Societal Impact:**

The authors do not specify the limitations of their model. It is true that their model seems pretty robust for their application, but some discussion about potential use in other problems would be appreciated, and it would make clear the limits of application of their approach.

Their contribution does not seem to imply any negative societal impact, thus it is not discussed.


**Main Review:**

The authors propose a novel approach to solve DPDP based on a bi-level structure which combines well-known techniques from reinforcement learning with Graph Isomorphism Networks to model the problem. The upper-level agent, which is responsible of the decision to wait for new requests, is what distinguishes their approach from previous ones, which are well explained and adequately cited. The authors claim to achieve superior performance over existing baselines, which is supported with an exhaustive experimental analysis on data generated by simulations based on real data. Results are confirmed to be consistent with further experimentation on real-time data.

Despite the authors study the ability of their approach to generalize to a different problem size, the ability to generalize to different distributions is not evaluated (data from the simulations and real-time data are likely to have the same distribution).

The paper is clearly written, except for small details listed here: (1) In section 3.2.1 it is not clear where remaining orders come from. (2) Section 3.3.1 does not make clear why reconstruction is needed after the lower-level agent reaches a local optimum. (3) In section 4.3 authors do not specify with respect to which measure the objective improvement has been computed in table 1. (4) In the same table the Overtime and Objective columns seem repetitive, since objective will only be different from average distance if there is some overtime, in which case the objective value does not bring any valuable information to the reader. Moreover, in table 1 the overtime is always 0 (and objective is always equal to average distance), so both columns could be avoided since they do not bring any information.

The approach proposed in the paper seems relevant to the community as it applies Reinforcement Learning approaches to an important and relevant problem (DPDP) improving on the state of the art. Researchers might use ideas from this paper to keep improving state of the art for DPDP or even for other similar fields, specially those which have some time-evolving component where the dynamics are not known a priori.

**************POST REBUTTAL COMMENTS*********************************

I read the rebuttals and they clarified some questions raised in the review.

In particular, the specific answer regarding generalization is interesting, but it does not address out-of-distribution generalization.
I agree this is a challenge but I would have appreciated a more detailed discussion on how this limits the current approach. As pointed out in the review, discussing potential application to other domains could clarify this point. However, authors in their answer essentially say that the model is general and could be used also in other problems without limitations. I think it is important to discuss the possible limitations of the approach rather than the possibility of applying it to other problems.

**Time Spent Reviewing:**

10

---

> ### Author Response · Authors · 2021-08-10
> **Response to Reviewer #4 (8RBm)**
>
> We thank Reviewer #4 (8RBm) for the valuable comments.
>
> **[About the evaluations on different distribution]**
>
> In the actual business scenario, orders generated in each day follow a similar distribution with a small variance. We have empirically verified that our method can bring improvement in offline experiments with data from the same distribution and in online experiments with data from similar distributions with small variations. We agree that addressing OOD (Out-of-distribution) problem is also important and worth further exploring which is currently a challenging problem in the overall machine learning community, but is out of the scope of this work.
>
> **[About the remaining orders]**
>
> The remaining orders refer to orders which have already been assigned to the vehicles but the goods of the orders are still not loaded onto the vehicles even at $t_{i+k}$. $t_{i+k}$ is the decision moment that the upper-level agent decides to release orders to the lower-level agent. Taking Figure 2 for example, we assume order A, B and C are assigned to a vehicle at decision moment $t_{1}$ (not drawn in the figure) and the vehicle starts to service order A and B first. However, the vehicle still has not started to service order C even at $t_{i+k}$. Then order C will become one of the remaining orders at $t_{i+k}$. We'll further explain it in our revised version.
>
> **[About the Reconstruction step]**
>
> Reconstruction is designed to help the lower-level agent to get rid of local optimum points where further improvement iterations are no longer improving the solutions. And also in our experiments, we have verified that this design helps find globally better solutions by starting a new improvement iteration with a reasonably good starting point. We'll further explain it in our revised version.
>
> **[About the objective improvement measure]**
>
> The objective improvement measurement is the improvement percentage of each algorithm compared with the Greedy algorithm. For example, the result of Greedy is 53.85 and the result of our method is 45.72, then the objective improvement of our method is calculated by -(45.72-53.85)/53.85 * 100%. We'll further explain it in our revised version.
>
> **[About the potential use in other problems]**
>
> Indeed our paper focuses on solving DPDP applications, while we would like to highlight that our proposed hierarchical RL framework is a quite general design and would be beneficial to researchers from ML and OR communities in general.
>
> Our framework can be potentially applied in a number of dynamic routing problems [2]. Dynamic routing problem is one of the most important problems in the area of enterprise logistics, including stochastic VRP, DPDP, online ride-hailing dispatch and so on. Apart from dynamic routing problems, our method can also be potentially applied in a number of other domains in the supply chain such as dynamic flow shop scheduling [3], dynamic job shop scheduling [4], dynamic bin packing [5] and so on. As orders/customers/tasks of all these dynamic problems are online generated and are not known a priori, the orders/customers/tasks should first be cached and then be dispatched. In this way, these problems can be formulated as hierarchical optimization problems like DPDP that the upper-level problem is ”how to cache orders/customers/tasks” and the lower-level problem is ”how to dispatch cached orders/customers/tasks”. Most existing approaches [2,3,4,5] partition the overall dynamic problems into static subproblems of fixed size/timespan and then solve each subproblem using manually designed heuristics. But these approaches are also faced with the same two major limitations as existing solutions for DPDP (detailed in literature review section):
> 1) The upper-level solvers myopically partition the dynamic problem into static sub-problems of fixed-size time window length, which restricts the method from finding a global better solution.
> 2) The lower-level solvers are normally heuristics-based, of which the design heavily relies on complex domain knowledge and is limited in generalization over different domains.
>
> As detailed in Section 3.1, to address these two kinds of limitations, we designed (1) an upper-level RL agent to optimize the long-term cumulative rewards to find a global better solution; (2) a lower-level RL agent to automatically generate effective and efficient heuristic search algorithms without complex manually design. Thus, we think that the core idea of our work might be beneficial to all the above problems in the supply chain community towards learning-based solutions.
>
> **[About the overtime column]**
>
> We'll omit the overtime column in Table 1 in our revised version as suggested.
>
> **Reference**
>
>     [1] Xijun Li, Weilin Luo, Mingxuan Yuan, Jun Wang, Jiawen Lu, Jie Wang, Jinhu Lv, and Jia Zeng. Learning to optimize industry-scale dynamic pickup and delivery problems. 37th IEEE International Conference on Data Engineering, 2021.
>     [2] Paolo Toth and Daniele Vigo. Vehicle Routing. Society for Industrial and Applied Mathematics, Philadelphia, PA, 2014.
>     [3] Lixin Tang, Wenxin Liu, and Jiyin Liu. A neural network model and algorithm for the hybrid flow shop154scheduling problem in a dynamic environment. Journal of Intelligent Manufacturing, 16(3):361–370, 2005.
>     [4] Jatoth Mohan, Krishnanand Lanka, and A Neelakanteswara Rao. A review of dynamic job shop scheduling techniques. Procedia Manufacturing, 30:34–39, 2019.
>     [5] Leah Epstein and Meital Levy. Dynamic multi-dimensional bin packing. Journal of discrete algorithms, 8(4):356–372, 2010.

---

### Official Review · Reviewer_JFQW · 2021-07-16

**Rating:** 6
**Confidence:** 4

**Summary:**

The current paper proposes a hierarchical approach to solve the dynamic pickup and delivery problem (DPDP). The method uses two RL modules. The upper-level RL policy is designed to segment the time window thus transforming DPDP into a static PDP. For the upper-level RL, DQN is used. The lower-level RL policy is trained to select one solution improving heuristics. The lover level policy is parameterized with the GNN network and trained under the framework of Reinforce.

**Limitations And Societal Impact:**

The authors did not comment on the limitation of the proposed method. The storing limitation is that the proposed approach is designed to tackle a very specific type of problem thus the proposed method cannot provide general insight or intuition for other related fields.

**Main Review:**

<Originality>
The current study aims to tackle the important and practical problems in the logistics and transportation fields. The paper explains well the importance of the DPDP. However, the proposed approach to solve the target problem is a combination of heuristics, mainly seeking to improve the performance rather than seeking to answer the fundamental research questions that can be rooted in the target problem.


The term “Bi-level optimization” is misleading. Bi-level optimization refers hierarchical optimization problem where the upper-level solution defines the lower-level optimization and thus affects the lower-level solution. In the proposed approach, it is difficult to understand how the upper-level and lower-level decision-making problems are interrelated. They just affect the final cost.

/


<Problem formulation>
I don’t quite understand what dynamic effect the upper-layer MDP trying to model. For example, what transition rule that DQN is trying to implicitly learn? Does the releasing action affect actually the long term performance? Can you verify if the estimated Q-value is actually modeling the future accumulated reward?

The action of the lower level of MDP is choosing one possible operator. It is not clear how such activities affect the state transition. In solving static PDP, there are numerous RL algorithms. I think the author should compare with one of these well-defined RL solvers for solving routing problems to justify the proposed method. In addition, the static PDF can be also solved with well know heuristics, such as LKH. In short, justify more the usage of a specific MDP formulation of the lower-level policy. In addition, there are many engineered heuristics in choosing a route or node to swap. Without learning selecting procedure for such operator-dependent actions, the learned policy is difficult to generalize over different sized problems, I think.

/

<Learning>
The learning method is not described in detail. How the upper-level and lower-level policies are trained? Are they trained simultaneously or alternatively?

/

<Baseline>
Comparing with the baseline using the fixed wind duration is not fair. Obviously using the variable window length is beneficial. The study should compare the different methods for segmenting the time window instead of comparing the fixed window and variable window.

/

<Clarity>
The paper is well written and well organized.

/

<Significance>
The proposed method proposed one of the possible heuristic approaches to solve DPDP. The proposed method rarely exploits the specific structure of the target problem and the proposed approach is far from tackling general research questions that the general ML community would be interested in. Thus, the significance is limited.



**Time Spent Reviewing:**

8 hours

---

> ### Author Response · Authors · 2021-08-10
> **Response to Reviewer #3 (JFQW)**
>
> We thank Reviewer #3 (JFQW) for the valuable comments.
>
> **[About comparing with RL algorithms on static PDP]**
>
> We compared DQN-based baselines both in our main experiments and ablation studies, and we are not sure which aspect of comparison the reviewer is referring
> :
> 1) If the reviewer means that we should compare with a method that the lower-level solver is an RL agent: In the main experiments of Section 4.3, we compare with 10min-Interval + ST-DDGN, which considers both the predicted orders and real generated orders when solving each sub-problem (high-level) and trains an end-to-end DQN agent to perform the order dispatching (low-level as described in Section 4.2).
> 2) If the reviewer means that we should compare with an RL method for static PDP in the ablation study: In Table 2 and Table 3 (Section 4.4.1) on static PDPs, we compare the end-to-end RL method with our approach.
>
> If the reviewer has any other particular baseline method in mind,  we’d also be very happy to provide the supplementary results by the end of the rebuttal phase.
>
> **[About LKH]**
>
> LKH and solving MIP using Gurobi can be classified as the same type of baselines because both of them are time-consuming methods (we found that LKH takes 4-12 hours, and Gurobi takes 8-54 hours on data set of Table 2), while the solution quality of Gurobi (i.e., the optimal solution) is usually better than LKH. The objective of the ablation study is to verify that our lower-level method can obtain high-quality approximated optimal solutions within an acceptable time. Therefore, we only compared our results with Gurobi and didn't present the results of LKH in the ablation study. We will clarify the reasons in the revised version.
>
> **[About generalization]**
>
> We train a shared policy network for the lower-level agent across different static PDP instances (of different scales). It takes the abstract features of each static PDP instance as input and outputs the probability of choosing different operators. The policy achieves generalization from the following three aspects:
> 1) $\textbf{State generalization}$: We incorporate GIN into our policy network to make it capable of handling input dimension change (the number of nodes is different over different scale problems) and the order change of input nodes (utilizing its permutation-invariant property). Thus the learned state representations can be generalized well over problems of different scales.
> 2) $\textbf{Action generalization}$: We fix the action set to the set of four operators to ensure the action space of the policy network is exactly the same over problems of different scales.
> 3) $\textbf{Completeness of the solution space}$: for any PDP instance, we have shown in Appendix G that any feasible solution (including the optimal one) could be obtained by iteratively applying the four operators from any given initial solution.
>
> Overall, the above three aspects of designs altogether ensure that the lower-level agent can be generalized well on problem instances of different scales.
>
> Apart from the methodology design, we have also empirically verified that our method generalizes well in Table 5 in Section 4.4.4. As shown in Table 5, the model trained on datasets of 300 orders achieves similar performances with the one trained on datasets of 1000 orders. Note that the model trained on datasets of 300 orders also outperforms the baselines in Table 1. The results showed that our method can be well generalized to new problems of different scales without fine-tuning after being well trained on existing problems.
>
> **[About learning details]**
>
> The upper-level agent is trained by minimizing loss in Equation 3 in the paper and the lower-level agent is trained using Equation 31 in the appendix. Both levels are trained simultaneously. This training process is relatively stable due to the following reasons. The iterative solution optimization process (starts from an initial greedy solution) of our lower-level agent described in Section 3.3.1 can ensure the obtained solutions have relatively high quality even at the initial training stages. In other words, the solutions given by the lower-level agent are relatively stable. Therefore, the unstable issue of co-training both levels of policies in our case is negligible, and thus both levels can be trained simultaneously. We will supplement and improve the explanation in our revised version.
>
> **[About baselines]**
>
> We agree that the influence of different variable time window strategies in upper-level policies is worth investigating. In the experiments, we compared with a variable time window method, i.e., ST-DDGN [1], which, to the best of our knowledge, is the current SOTA method. ST-DDGN predicts future orders of DPDP and both the predicted orders and real generated orders are considered when solving each sub-problem. Predicting future orders can be viewed as a variant of dynamically segmenting the time window, since the upper-level agent considers both the orders accumulated in the current 10min-Interval and the predicted orders that would appear in the subsequent 10min-Intervals.
>
> **[About the societal impact and the potential use in other problems]**
>
> Indeed our paper focuses on solving DPDP applications, while we would like to highlight that our proposed hierarchical RL framework is a quite general design and would be beneficial to researchers from ML and OR communities in general.
>
> Our framework can be potentially applied in a number of dynamic routing problems [2]. Dynamic routing problem is one of the most important problems in the area of enterprise logistics, including stochastic VRP, DPDP, online ride-hailing dispatch and so on. Apart from dynamic routing problems, our method can also be potentially applied in a number of other domains in the supply chain such as dynamic flow shop scheduling [3], dynamic job shop scheduling [4], dynamic bin packing [5] and so on. As orders/customers/tasks of all these dynamic problems are online generated and are not known a priori, the orders/customers/tasks should first be cached and then be dispatched. In this way, these problems can be formulated as hierarchical optimization problems like DPDP that the upper-level problem is ”how to cache orders/customers/tasks” and the lower-level problem is ”how to dispatch cached orders/customers/tasks”. Most existing approaches [2,3,4,5] partition the overall dynamic problems into static subproblems of fixed size/timespan and then solve each subproblem using manually designed heuristics. But these approaches are also faced with the same two major limitations as existing solutions for DPDP (detailed in literature review section):
> 1) The upper-level solvers myopically partition the dynamic problem into static sub-problems of fixed-size time window length, which restricts the method from finding a global better solution.
> 2) The lower-level solvers are normally heuristics-based, of which the design heavily relies on complex domain knowledge and is limited in generalization over different domains.
>
> As detailed in Section 3.1, to address these two kinds of limitations, we designed (1) an upper-level RL agent to optimize the long-term cumulative rewards to find a global better solution; (2) a lower-level RL agent to automatically generate effective and efficient heuristic search algorithms without complex manually design. Thus, we think that the core idea of our work might be beneficial to all the above problems in the supply chain community towards learning-based solutions.
>
> **Reference**
>
>     [1] Xijun Li, Weilin Luo, Mingxuan Yuan, Jun Wang, Jiawen Lu, Jie Wang, Jinhu Lv, and Jia Zeng. Learning to optimize industry-scale dynamic pickup and delivery problems. 37th IEEE International Conference on Data Engineering, 2021.
>     [2] Paolo Toth and Daniele Vigo. Vehicle Routing. Society for Industrial and Applied Mathematics, Philadelphia, PA, 2014.
>     [3] Lixin Tang, Wenxin Liu, and Jiyin Liu. A neural network model and algorithm for the hybrid flow shop154scheduling problem in a dynamic environment. Journal of Intelligent Manufacturing, 16(3):361–370, 2005.
>     [4] Jatoth Mohan, Krishnanand Lanka, and A Neelakanteswara Rao. A review of dynamic job shop scheduling techniques. Procedia Manufacturing, 30:34–39, 2019.
>     [5] Leah Epstein and Meital Levy. Dynamic multi-dimensional bin packing. Journal of discrete algorithms, 8(4):356–372, 2010.

---

> > ### Comment · Reviewer_JFQW · 2021-08-27
> > **Thank you for clarifying.**
> >
> > The authors have well addressed my concen. It would be better to clearly discuss how the upper and lower level problems are interrelated in the main text. I also recommend providing the overall formulation of the target problem. I increase my evaluation score from 5 to 6.

---

> > > ### Author Response · Authors · 2021-09-02
> > > **Follow-up response to Reviewer #3 （JFQW)**
> > >
> > > Thank you for your valuable comments. We will continue to polish our paper as your suggestions.

---

> ### Author Response · Authors · 2021-08-10
> **Response to Reviewer #3 (JFQW)**
>
> We thank Reviewer #3 (JFQW) for the valuable comments.
>
> **[About interrelations of upper-level and lower-level problems]**
>
> To address your concern about the interrelations of upper-level and lower-level problems, we first give a formulation of the DPDP problem below.
> $$
> \begin{aligned}
> \max_{\Pi} \max_{\mathcal{X}} \sum_{i=1}^{N} x_{i} \& V_{\pi_{i}}\left(\text{ro}\_{j}, \text{no}\_{j \rightarrow i}, \text{vs}\_i\right) \\\\
> \& x_{j}=1 \text { and } j=\max \\{1,2, \ldots, i-1\\}
> \end{aligned}
> $$
> where $N$ is the total numbers of decision making times of the upper-level agent, $\\mathcal{X}=\\left\\{x\_{1}, \\ldots, x\_{N}\\right\\}$, $x\_{i} \\in \\{0,1\\}$ indicates whether the upper-level agent releases orders to the lower-level agent. $\\Pi=\\left\\{\\pi\_{1}, \\ldots, \\pi\_{N}\\right\\}$, and $\\pi\_{i}$ represents the control policy of the lower-level agent for solving subproblem $i$.
>
> If $x_{i}=1$, a new subproblem $i$ (i.e., static PDP) is released. This newly generated subproblem is mainly defined upon three key components which are directly influenced by the previous upper-level solution.
> 1) $\\text{ro}\_{j}$: the Remaining (undelivered) Orders of the previous released subproblem $j$ ($x\_{j}=1 \\text { and } j=\\max \\{1,2, \\ldots, i-1\\} $).
> 2) $\\text{no}\_{j \\rightarrow i}$: Newly generated Orders between timestep $j$ and $i$. Different $j$ will result in different $\\text{ro}\_{j}$ and $\\text{no}\_{j \\rightarrow i}$.
> 3) $\\text{vs}\_i$: the Vehicle Status at decision point $i$. Note that $\\text{vs}\_i$ is a direct result of the dispatching solutions of previous generated subproblems which is controlled by $\\left\\{x\_{1}, \\ldots, x\_{i-1}\\right\\}$.
>
> Thus, the lower-level optimization is directly defined by the upper-level solution. Based on these 3 components, the objective function of subproblem $i$ under policy $\\pi\_{i}$ is denoted as $V\_{\\pi\_{i}}\\left(\\text{ro}\_{j}, \\text{no}\_{j \\rightarrow i}, \\text{vs}\_i\right)$.
>
> Besides, it's worth noting that the lower-level solution will also influence the upper-level decision-making. Specifically, different lower-level policies $\\pi\_i$ at current timestep $i$ will lead to different dispatching results, e.g., different $\\text{ro}\_i$s and $\text{vs}\_{i+1}$s, which are the main components of the next subproblem $i+1$, and thus will influence timestep $i+1$’s decision making, i.e., we should set $x\_{i+1}$ to $0$ or $1$.
>
> Therefore, the upper-level and lower-level decision-making problems are interrelated. But due to the characteristics of online and sequential decision-making, this formulation of DPDP is still different from the definition of the standard bilevel optimization problem. As also pointed out by Reviewer #2 (T5ZV), we agree that it’s more accurate to denote our problem definition as a hierarchical optimization problem. We will rename it in the revised version.
>
> **[About dynamic effect the upper-layer MDP trying to model]**
>
> As described in the previous formulation, the state of the upper-level MDP mainly consists of three components: $\\text{ro}\_{j}$, $\\text{no}\_{j \\rightarrow i}$ (the status of the remaining and generated orders, e.g., the number and overtime risk of these orders) and $\\text{vs}\_i$. If high-level action $x_i$ is set to 0 (do not release subproblem $i$), the orders cached in the buffer will be further accumulated, i.e., from $\\text{no}\_{j \\rightarrow i}$ to $\\text{no}\_{j \\rightarrow i+1}$. Otherwise, if $x_i$ is set to 1, a new subproblem $i$ is released to the lower-level agent, and the upper-level state transits to $\\text{ro}\_{i}$, $\\text{no}\_{i \\rightarrow i+1}$, $\\text{vs}\_{i+1}$ at the next decision point $i+1$.
>
> In general, setting $x_i=0$ would lead to accumulating more orders and the overall travelling distances could be potentially optimized shorter (since each vehicle will have more candidate orders to choose), but at the risk of the increase of the overtime for earlier cached orders, and vice versa. Thus, different upper-level policies would lead to different overall traveling distances and overtime, which is exactly our optimization objective, i.e., minimizing the expected long-term cumulative distances and overtime. We’ll further clarify it in the revised version.

---

### Official Review · Reviewer_T5ZV · 2021-07-21

**Rating:** 7
**Confidence:** 3

**Summary:**

The paper considers dynamic pickup and delivery problems in which a limited number of vehicles need to be scheduled to handle all demands with the goal of minimizing the average traveling distance. The problem involves capacity constraints of the vehicles, LIFO constraint, and soft time window constraint, where violating the time windows result in some penalty.

A new algorithm is proposed which is based on the existing idea: Make a buffer to cache the newly arrived orders, then periodically dispatch all the orders in the buffer. The current approaches add the orders to the buffer up to a given size or based on the prediction of the future, but the new proposed approach uses an upper-level agent which determines whether to wait for more items (and get some delay penalty) or dispatch the current orders. For this purpose, a day is partitioned into 144 time-slots of each 10 minutes and the goal is to decide to continue or wait at each of those steps. Then, the received ordered in the buffer with all the orders that are not picked up will be passed to the lower-level agent to solve the static pick-up and delivery sub-problem. The upper-level agent uses DQN to train the agent in which the same goal as of the problem is used as the reward. The state includes "the number of orders accumulated in the buffer, the number of available vehicles, the amount of time left before exceeding the time limit of each order".
The lower level agent learns the assignment of the orders to the vehicles and the sequence to pass over the orders. The state includes the position information of the customers and vehicles. Action is choosing one of four introduced heuristic improvements for routes. They can be applied to the routes and if there are any improvements, the new route will be accepted. A graph neural network with REINFORCE algorithm is used to train the policy.


**Ethics Review Area:**

["I don’t know"]

**Limitations And Societal Impact:**

There is no social impact and the limitations of the work are not discussed.

**Main Review:**

Originality: A new method based on hierarchical RL is used to address an existing problem.  Compared to the previous papers, the contributions are clear in terms of the model and the results. I am not aware of all the recent research in this field, but as long as my limited knowledge allows, there are several recent non-RL algorithms for this problem which are not cited.

Quality: The submission technically sound? The claims are supported by experimental results. The weaknesses and the limitation of the work are not discussed.

Clarity: The submission is clearly written and it is organized OK.

Significance: The results are important and probably the results will be used by other researchers or practitioners. It seems that it does advance the state of the art based on numerical experiments. It provides an environment based on a real-world dataset.

I enjoyed reading the paper. I have few comments:


Q1- In several places of the paper the term "bi-level reinforcement learning (RL) based optimization framework" is used. This is misleading since bi-level optimization is a known type of problem in mathematical programming in which the solution of an optimization problem is a constraint in the main problem (see the problem definition and the formulation in [1] as a recent paper in this field). This is not the case in the problem that you have defined. The algorithm that is defined, has two levels, which are commonly named hierarchical RL. I suggest revising the paper accordingly to avoid any confusion in the future.

Q2- Several recent papers are not missed in the literature review section. I would add and review the contributions of [3, 4, 5, 6]. For older papers see [2].

Minor comments:
"Although these methods mainly focus on TSPs or VRPs, of which all orders’ information is known in advance and much fewer constraints are considered comparing with DPDP, learning-based methods have shown great potential to help solve large-scale DPDPs and reach superior performance."
This is not correct. There are several works on RL for VRP which consider stochastic VRP problems in which the demands and their locations are not known a priori.

[1] Tahernejad, S., Ralphs, T.K. & DeNegre, S.T. A branch-and-cut algorithm for mixed integer bilevel linear optimization problems and its implementation. Math. Prog. Comp. 12, 529–568 (2020). https://doi.org/10.1007/s12532-020-00183-6

[2] Berbeglia, Gerardo, Jean-François Cordeau, and Gilbert Laporte. "Dynamic pickup and delivery problems." European journal of operational research 202, no. 1 (2010): 8-15.

[3] Karami, Farzaneh, Wim Vancroonenburg, and Greet Vanden Berghe. "A periodic optimization approach to dynamic pickup and delivery problems with time windows." Journal of Scheduling 23, no. 6 (2020): 711-731.

[4] Ulmer, Marlin W., Barrett W. Thomas, Ann Melissa Campbell, and Nicholas Woyak. "The restaurant meal delivery problem: Dynamic pickup and delivery with deadlines and random ready times." Transportation Science 55, no. 1 (2021): 75-100.

[5] Su, Zhiyuan, Wantao Li, Jicao Li, and Bin Cheng. "Heterogeneous fleet vehicle scheduling problems for dynamic pickup and delivery problem with time windows in shared logistics platform: formulation, instances and algorithms." International Journal of Systems Science: Operations & Logistics (2021): 1-25.

[6] Györgyi, Péter, and Tamás Kis. "A probabilistic approach to pickup and delivery problems with time window uncertainty." European Journal of Operational Research 274, no. 3 (2019): 909-923.



**Time Spent Reviewing:**

6

---

> ### Author Response · Authors · 2021-08-10
> **Response to Reviewer #2 (T5ZV)**
>
> We thank Reviewer #2 (T5ZV) for the valuable comments.
>
> **[About Q1]**
>
> We thank the reviewer for raising the question about the unclarity of the term “bi-level optimization”. After comparing our DPDP formulation with the standard bilevel optimization formulation of the recommended paper [1], we agree that it’s more accurate to name our method as a “hierarchical RL optimization framework” instead of “bi-level optimization”. We’ll modify our statements in the revised version.
>
> **[About Q2]**
>
> We thank the reviewer for bringing the works [2-6] into our attention and we carefully read the recommended papers. These non-RL algorithms [3-6] for DPDP belong to the first category we listed in the literature review part. We will cite these algorithms and the survey paper [2] properly in the revised version.
>
> **[About minor comments]**
>
> Thank you for the valuable comments for the paper presentation and we will revise our statements carefully.
>
> **Reference**
>
>     [1] Sahar Tahernejad, Ted K Ralphs, and Scott T DeNegre. A branch-and-cut algorithm for mixed integer bilevel linear optimization problems and its implementation. Mathematical Programming Computation, 12(4):529–568, 2020.
>     [2] Gerardo Berbeglia, Jean-François Cordeau, and Gilbert Laporte. Dynamic pickup and delivery problems. European journal of operational research, 202(1):8–15, 2010.
>     [3] Farzaneh Karami, Wim Vancroonenburg, and Greet Vanden Berghe. A periodic optimization approach to dynamic pickup and delivery problems with time windows.Journal of Scheduling, 23(6):711–731, 2020.
>     [4] Marlin W Ulmer, Barrett W Thomas, Ann Melissa Campbell, and Nicholas Woyak. The restaurant meal delivery problem: Dynamic pickup and delivery with deadlines and random ready times. Transportation Science, 55(1):75–100, 2021.
>     [5] Zhiyuan Su, Wantao Li, Jicao Li, and Bin Cheng. Heterogeneous fleet vehicle scheduling problems for dynamic pickup and delivery problem with time windows in shared logistics platform: formulation, instances and algorithms. International Journal of Systems Science: Operations & Logistics, pages 1–25, 2021.
>     [6] Péter Györgyi and Tamás Kis. A probabilistic approach to pickup and delivery problems with time window uncertainty. European Journal of Operational Research, 274(3):909–923, 2019.

---

### Official Review · Reviewer_WBgW · 2021-07-22

**Rating:** 7
**Confidence:** 3

**Summary:**

The paper proposed a framwork to solve  Dynamic Pickup and Delivery Problem (DPDP) via Reinforcement Learning Based Bi-level
Optimization method. The upper-level agent used DQN to make a decision whether to release cached orders, while the lower-level agent used GNN to solve each sub-problem. The overall objective function is the weighted sum of travelling distances and overtime. This proposed framwork got a superior performance over baseline.

**Ethical Concerns:**

no ethical issues

**Limitations And Societal Impact:**

Yes

**Main Review:**

Strengths:
The paper applied the idea of bi-level optimization to DPDP, which is a novel solution for DPDP. And the detailed algorithm of each level is also given.
The upper-level uses DQN to eliminate the need to maintain a fixed size cache or prediction for future orders.
The experimental results show that this method is worth popularizing

Weaknesses:
It seems to be a technical application paper, rather than an academic research one. So the overall novelty may be limited but still interesting. Also, I have a feeling that the training process might be unstable or hyperparameters may be sensitive?

**Time Spent Reviewing:**

3

---

> ### Author Response · Authors · 2021-08-04
> **Unmatched review**
>
> Dear PCs, SACs, ACs and reviewer WBgW:
>
> After carefully reading the reviews from Reviewer WBgW, we found the reviewer submit a comment that does not match our paper. It seems the reviewer accidentally submitted the review of other papers as the review of our paper.
>
> In the reviewer's summary, he/she said 'This paper proposed a framework, ClassMate, to improve BLO-based models via inspiration from humans’ classroom study techniques. Also, it applied ClassMate to NAS and data weighting by emulating the Feynman technique and peer questioning.' However, our paper focus on solving Dynamic Pickup and Delivery Problem instead of BLO-based model and our framework name is not ''ClassMate'.  And other details in Main Review also do not match our paper.
>
> Could you please help check the reviews?

---

> > ### Comment · Reviewer_WBgW · 2021-08-04
> > **Very sorry for my carelessness**
> >
> > I'm terribly sorry for my carelessness！I opened multiple web interfaces at that time, and then accidentally posted another review to here.
> > I have corrected my mistake and given my review comments at the time. The authors proposed a very noval solution for DPDP and did a good job. I appreciate it very much.
> > Once again, I sincerely apologize for my mistake!

---

> ### Author Response · Authors · 2021-08-10
> **Response to Reviewer #1 (WBgW)**
>
> We thank Reviewer #1 (WBgW) for the valuable comments.
>
> **[About the training process]**
>
> Thank you for pointing out this question. Indeed, in general the training process of a hierarchical RL would be unstable if both the upper-level and lower-level agents are trained simultaneously from scratch. A typical way to alleviate the training instability is to periodically iterate the training of lower-level and upper-level agents, i.e., train the lower-level agent firstly for a while and then update the upper-level agent based on the partially trained lower-level policy and finally train both levels simultaneously until convergence. However, in our case, the iterative solution optimization process (starts from an initial greedy solution) of our lower-level agent can ensure the obtained lower-level solutions have relatively high quality even at the initial training stages, as described in Section 3.3.1. In other words, the solutions given by the lower-level agent are relatively stable. Therefore, the unstable issue of co-training both levels of policies in our case is negligible, and thus both levels can be trained simultaneously. It can also be empirically shown in Figure 11 of the Appendix that the learning process (training curve in red）of our approach is relatively stable.
>
> **[About the hyperparameters]**
>
> We performed experiments under various hyperparameters and verified our method is not sensitive to hyperparameters. The hyperparameters of our method are mainly constituted of two classes: 1) the hyperparameters of GIN architectures; 2) the hyperparameter settings of RL algorithms. For each class of hyperparameters, a number of candidate hyperparameters are generated based on the settings of GIN[1], DQN[2] and REINFORCE[3]. Then, we got the optimal hyperparameters via grid search from these candidate hyperparameters and presented the corresponding results in our paper. During grid search, we found our method is robust under hyperparameter variations. We will further explain this in the revised version.
>
> **Reference**
>
>     [1] Keyulu Xu, Weihua Hu, Jure Leskovec, and Stefanie Jegelka. How Powerful are Graph Neural Networks?. In International Conference on Learning Representations, 2019.
>     [2] Volodymyr Mnih, Koray Kavukcuoglu, David Silver, Andrei A Rusu, Joel Veness, Marc G Bellemare, Alex Graves, Martin Riedmiller, Andreas K Fidjeland, Georg Ostrovski, et al. Human-level control through deep reinforcement learning. nature, 518(7540):529–533, 2015.
>     [3] Ronald J Williams. Simple statistical gradient-following algorithms for connectionist reinforcement learning. Machine learning, 8(3-4):229–256, 1992.
>     [4] Paolo Toth and Daniele Vigo. Vehicle Routing. Society for Industrial and Applied Mathematics, Philadelphia, PA, 2014.
>     [5] Lixin Tang, Wenxin Liu, and Jiyin Liu. A neural network model and algorithm for the hybrid flow shop154scheduling problem in a dynamic environment.Journal of Intelligent Manufacturing, 16(3):361–370, 2005.
>     [6] Jatoth Mohan, Krishnanand Lanka, and A Neelakanteswara Rao. A review of dynamic job shop scheduling techniques. Procedia Manufacturing, 30:34–39, 2019.
>     [7] Leah Epstein and Meital Levy. Dynamic multi-dimensional bin packing. Journal of discrete algorithms, 8(4):356–372, 2010.

---

> > ### Comment · Reviewer_WBgW · 2021-08-25
> > **I'm inclined to accept.**
> >
> > Thank you for explaining the question I raised.  I think it is an interesting job and I'm inclined to accept.

---

> > > ### Author Response · Authors · 2021-09-02
> > > **Follow-up response to Reviewer #1 (WBgW)**
> > >
> > > Thank you for your approval. We will continue to polish our paper as your suggestions.

---

### Decision · Program_Chairs · 2021-09-27

**Decision:**

Accept (Poster)

**Comment:**

All reviewers are positive to this paper. The pickup and deliver problem is important and the bi-level optimization solution (or hierarchical RL) shows nontrivial technical depth.

Most concerns or confusions have been addressed or clarified by the rebuttal and the multiple round in the interaction. This paper is recommended to be accepted. Please follow the post review to appropriately polish this paper, for example, the generalization issue needs to be discussed in the revision.